# A Unifying Framework of Off-Policy General Value Function Evaluation

**Tengyu Xu**
Meta Platforms, Inc
Menlo Park, CA 94025

**Zhuoran Yang**
Yale University
New Haven, CT 06520

**Zhaoran Wang**
Northwestern University
Evanston, IL 60208

**Yingbin Liang**
Ohio State University
Columbus, OH 43210

## Abstract

General Value Function (GVF) is a powerful tool to represent both the *predictive* and *retrospective* knowledge in reinforcement learning (RL). In practice, often multiple interrelated GVFs need to be evaluated jointly with pre-collected off-policy samples. In the literature, the gradient temporal difference (GTD) learning method has been adopted to evaluate GVFs in the off-policy setting, but such an approach may suffer from a large estimation error even if the function approximation class is sufficiently expressive. Moreover, none of the previous work have formally established the convergence guarantee to the ground truth GVFs under the function approximation settings. In this paper, we address both issues through the lens of a class of GVFs with causal filtering, which cover a wide range of RL applications such as reward variance, value gradient, cost in anomaly detection, stationary distribution gradient, etc. We propose a new algorithm called GenTD for off-policy GVFs evaluation and show that GenTD learns multiple interrelated multi-dimensional GVFs as efficiently as a single canonical scalar value function. We further show that unlike GTD, the learned GVFs by GenTD are guaranteed to converge to the ground truth GVFs as long as the function approximation power is sufficiently large. To our best knowledge, GenTD is the first off-policy GVF evaluation algorithm that has global optimality guarantee.

## 1 Introduction

The value function, which represents the expected accumulation of reward [43], serves as a reliable performance metric of policy in the reinforcement learning (RL) tasks [42, 23]. In many RL applications, however, looking at only the value function is not enough. For example, in the risk-sensitive domains such as health care and financial assets, the variance of "reward-to-go" rather than the value function, i.e., the mean of "reward-to-go", is a more suitable performance metric. As another example, to obtain a variance-reduced or bias-reduced policy gradient estimator [14, 61, 18], in addition to the value function, the information of "gradient of value function" is also required. Moreover, in continuous control domain with differentiable and deterministic policy, the computation of policy gradient is only possible through "action/state-value gradient" [38, 8, 12], etc. All the aforementioned metrics can be viewed as *predicative* knowledge of certain multiple intercorrelated cumulative "signals" (possibly high-dimensional, e.g., the gradient of value function), and thus naturally fall into the framework of **forward GVFs** (refers to forward general value functions) [48, 57, 32, 33] (see Section 2.1 for the formal definition). One typical approach to evaluate GVFs, is to learn from samples that pre-collected from one or more behavior policies, which yields an

*off-policy* method. In practice, multiple forward GVFs are usually evaluated jointly at the same time due to their interrelationships [37, 33].

In contrast to forward GVFs defined based on predictive knowledge, the **backward GVFs** represents *retrospective* knowledge, which captures the accumulation of signals from the past to the present time [70] (see Section 2.2 for the formal definition). Although the concept of the backward GVFs has not been formally proposed until very recently [70], it is rooted in a number of important RL applications such as anomaly detection [70], emphatic weight learning [46, 70] and evaluation of gradient of logarithmic stationary distribution [26, 61, 18]. Different from the forward GVFs, for which the Bellman operator can be defined independently from the sampling distribution [37, 42, 44], the Bellman operator of the backward GVFs is only valid if the sampling exactly follows the on-policy stationary distribution [70]. Due to such a reason, *off-policy* evaluation of the backward GVFs is much more challenging than that of the forward GVFs.

In general, due to the high dimensionality and intercorrelation, it is very challenging to evaluate multiple GVFs simultaneously with standard policy evaluation approaches [55, 45, 23]. In previous studies, the gradient temporal difference (GTD) learning [45, 23], one of the most popular off-policy methods in value function evaluation, has been adopted to solve both the forward and backward GVFs evaluation problems [48, 37, 69]. GTD adopts the mean squared projected Bellman error (MSPBE) as its optimization objective and takes the expectation over the *behavior* policy, which does not exactly reflect the desirable evaluation under the *target* policy. As a result, GTD can encounter serious issues in GVFs evaluation problems. **First**, the optimal point to which GTD converges can be far away from the ground truth value of GVFs. It becomes worse when multiple GVFs are evaluated simultaneously, because the error of one GVFs evaluation can be further amplified across other GVFs' evaluation due to their inherent correlations. In the literature, no provable bound has been established on such an error, which can, in fact, be unbounded for some cases (see Example 1 in [19]). **Second**, for high-dimensional GVFs evaluations, the landscape geometry of the GTD objective function can be ill-conditioned [23], which could slow down the convergence of GTD significantly. As demonstrated by our empirical results in Section 5, GTD can suffer from both the large estimation error and the slow convergence rate, which further suggests that GTD may not be a good choice for GVFs evaluation tasks. This motivates our paper to address the following question:

- *Can we design a new off-policy approach for multiple interrelated and high-dimensional GVFs evaluation problems, which is guaranteed to converge fast and converge to the ground truth GVFs?*

**Our Contributions.** In this paper, we investigate the problem of evaluating multiple interrelated GVFs jointly. Rather than studying different GVFs on a case-by-case basis, we explore the class of "GVFs with causal filtering", which captures a common structural feature shared by GVFs in a wide range of RL applications (see Appendix C). **(a)** We prove that both forward and backward GVFs with causal filtering are the unique fixed point of their corresponding general Bellman operator (GBO) (defined for multiple high-dimensional GVFs), which is shown to have a contraction property with respect to a properly constructed norm metric. **(b)** Based on such a property of GVFs, we propose a new algorithm GenTD to solve off-policy GVFs evaluation problem. GenTD introduces a density ratio to adjust the behavior distribution and further incorporates a policy-agnostic approach GenDICE/GradientDICE [65, 70] for estimating the density ratio jointly with GVF evaluation. **(c)** In the linear function approximation setting, we show that GenTD converges to the globally optimal point at the rate of $\mathcal{O}(1/T)$, with conditional number independent from the dimension of GVFs. Such a result implies that GenTD learns multiple interrelated possibly high-dimensional GVFs as efficiently as TD learning for a single canonical scalar value function. **(d)** We further show that unlike GTD, GenTD is guaranteed to approximate the ground truth GVFs well as long as the function expressive power is sufficiently large. To our best knowledge, GenTD is the first off-policy GVF evaluation algorithm that has such a ground truth guarantee. **(e)** Our experiments further demonstrate that GenTD converges much faster than GTD, and more importantly, converges to ground truth closely, whereas GTD suffers from large approximation error.

**Related Work.** The forward GVF was first introduced in [48] to represent a set of accumulation of general signals with possibly time-varying discount factors. The forward GVF was later used to represent a set of interrelated predictions [37, 9, 41, 24, 33]. It has been observed that some RL metrics such as variance, gradient of value function, state/action value gradient can also be viewed as forward GVFs [51, 14, 61, 18, 38, 48, 57, 32, 8, 5]. In previous works, both TD learning and GTD have been used to evaluate forward GVFs in the on- and off-policy settings [48, 37, 33], respectively.

A more comprehensive review of studies of forward GVFs has been provided in [35]. The backward GVF was formally defined in [70]. Some previous works have also considered metrics that can be represented as accumulations of signals in the reverse time direction, such as emphatic weighting, page ranking cost, and derivative of logarithmic stationary distribution [66, 68, 26, 63, 10]. Another track of research has focused on evaluation of a general scalar function in the off-policy setting, a.k.a off-policy evaluation (OPE) [4, 64, 16]. However, since the focus of this paper is on the evaluation of multiple high-dimensional GVFs, results in OPE are not directly comparable with ours.

The theoretical studies of off-policy GVFs evaluation algorithms are rather limited. So far, only the asymptotic convergence guarantee (without the convergence rate characterization) of GTD has been established in both the forward and backward GVFs evaluation settings [37, 70]. The convergence rate of GTD has only been established in [62, 7, 17, 6, 58, 21] for the simple canonical value function evaluation setting, which is a special case of forward GVFs. However, as pointed out in [19, 10, 27], the optimal point of GTD may suffer from possibly unbounded approximation error, which is not desirable in practice. In contrast, we propose a new off-policy GVFs evaluation algorithm, which can solve a wide range of forward and backward GVFs evaluation problems, with convergence rate characterization and guaranteed optimality with respect to the ground truth GVFs value.

We note that the contraction property of GBO for forward GVFs has also been investigated in [33] with a structure called "acyclic graph", which is similar to "causal filtering" in our paper (see the footnote comment for Proposition 1). However, the results of backward GVFs are established in our work for the first time. The focus of this paper is on the finite time performance and optimality guarantee for a new off-policy GVFs evaluation algorithm GenTD, which was not studied in [33].

## 2 Markov Decision Process and General Value Function

We consider an infinite-horizon Markov Decision Process (MDP) with a state space $\mathcal{S}$, an action space $\mathcal{A}$, a reward function $r : \mathcal{S} \times \mathcal{A} \to \mathbb{R}$, a transition kernel $\mathsf{P} : \mathcal{S} \times \mathcal{S} \times \mathcal{A} \to [0,1]$, a discounted factor $\gamma \in (0,1)$, and an initial distribution $\mu_0 : \mathcal{S} \to [0,1]$. An policy $\pi(a|s)$ is the probability of taking action $a$ at state $s$. At time step $t$, an agent at a state $s_t$ selects an action $a_t$ according to $\pi(\cdot|s_t)$, receives a reward $r(s_t, a_t)$, and transits to state $s_{t+1}$ according to $\mathsf{P}(\cdot|s_t, a_t)$. The state-action transition kernel is defined as $\mathsf{P}_\pi \in \mathbb{R}^{|\mathcal{S}||\mathcal{A}| \times |\mathcal{S}||\mathcal{A}|}$, in which $\mathsf{P}_\pi((s,a),(s',a')) = \mathsf{P}(s'|s,a)\pi(a'|s')$. When the MDP is ergodic, we define $\mu_\pi$ as the state-action stationary distribution which satisfies: $\mu_\pi^\top \mathsf{P}_\pi = \mu_\pi^\top$. For such an MDP, we define the discounted accumulation of reward as the "reward-to-go": $J_\pi = \sum_{t=0}^\infty \gamma^t r(s_t, a_t)$. The state-action value function (i.e., Q-function) is defined as $Q_\pi(s,a) = \mathbb{E}[J_\pi|(s_0, a_0) = (s,a)]$, and the state value function (i.e., V-function) is defined as $V_\pi(s) = \mathbb{E}[Q_\pi(s,a)|s]$. Note that $Q_\pi(s,a)$ satisfies the following Bellman equation

$$Q_\pi = \mathcal{T}_\pi Q_\pi = R + \gamma \mathsf{P}_\pi Q_\pi, \tag{1}$$

where $\mathcal{T}_\pi$ is the Bellman operator, and $Q_\pi$ and $R \in \mathbb{R}^{|\mathcal{S}||\mathcal{A}|}$ are vectors obtained via stacking $Q_\pi(s,a)$ and $r(s,a)$ over state-action space $\mathcal{S} \times \mathcal{A}$. We introduce a function of $(s,a)$ (possibly in the vector form) as $v(s,a) \in \mathbb{R}^d$ ($d \geq 1$). Consider a distribution $\xi(\cdot)$ over $\mathcal{S} \times \mathcal{A}$. We define the $\xi$–norm of $v \in \mathbb{R}^{d|\mathcal{S}||\mathcal{A}|}$ as $\|v\|_\xi = \sqrt{\sum_{(s,a)} \xi(s,a) \|v(s,a)\|_2^2}$, where $v$ is obtained by stacking the function $v(s,a)$ over $\mathcal{S} \times \mathcal{A}$. It has been proved that $\mathcal{T}_\pi$ is $\gamma$–contraction in $\mu_\pi$–norm, i.e., $\|\mathcal{T}_\pi v - \mathcal{T}_\pi v'\|_{\mu_\pi} \leq \gamma \|v - v'\|_{\mu_\pi}$ and $Q_\pi$ is the unique fixed point of $\mathcal{T}_\pi$ [44, 42, 55]. In the sequel, we denote $\mathsf{I}_d$ as the identity matrix with the dimension $d$ and $\otimes$ as the Kronecker product. We further define $U_\pi = \text{diag}(U_{\pi,1}, \cdots, U_{\pi,k})$, in which $U_{\pi,i} = \text{diag}(\mu_\pi) \otimes \mathsf{I}_{d_i}$ for $i = \{1, \cdots, k\}$, and $P_\pi = \text{diag}(P_{\pi,1}, \cdots, P_{\pi,k})$, in which $P_{\pi,i} = \mathsf{P}_\pi \otimes \mathsf{I}_{d_i}$.

### 2.1 Forward General Value Function

Consider a set of the state-action general value functions (GVFs) $G_\pi = [G_{\pi,1}^\top, \cdots, G_{\pi,k}^\top]^\top$, where each GVF $G_{\pi,i}$ is defined as the accumulation of a corresponding signal $C_i(s,a) \in \mathbb{R}^{d_i}$ given by $G_{\pi,i}(s,a) = \mathbb{E}\left[\sum_{t=0}^\infty \gamma_i^t C_i(s_t, a_t) \big| (s_0, a_0) = (s,a), \pi\right]$, where $\gamma_i \in (0,1)$ is a discount factor associated with $C_i$. Since $C_i(s,a) \in \mathbb{R}^{d_i}$ can be high-dimensional, $G_{\pi,i}(s,a)$ can also be high-dimensional for each $(s,a)$. Clearly, the Q-function is a special GVF associated with a scalar signal. Since $G_\pi$ is defined as the accumulation of the signal $C_i$ in a forward direction from the current time step $t$ to the future $\infty$, we call $G_\pi$ as "forward GVF".

In many RL applications, we are often interested in the case that $G_\pi$ of GVFs have progressive dependence [48], i.e., each $C_i(s,a)$ (associated with $G_{\pi,i}$) depends on the lower-indexed value functions $G_{\pi,1}, \cdots, G_{\pi,i-1}$ in the set. As a concrete example, suppose the policy is parametrized by a smooth function $\pi_w$, where the parameter $w \in \mathbb{R}^{d_w}$. In addition to the Q-function $Q_\pi$, the gradient $\nabla_w Q_\pi(s,a)$ of the Q-function w.r.t $w$ arises as a GVF of interest in several applications. In such a case, $G_{\pi,1} = Q_\pi$ and $G_{\pi,2} = \nabla_w Q_\pi$. It has been shown in [5] that the reward $C_2(s,a)$ associated with $\nabla_w Q_\pi$ is given by $C_2(s,a) = \gamma \mathbb{E}[Q_\pi(s',a')\nabla_w \log(\pi_w(s',a'))|s,a]$, which depends on the lower-indexed $G_{\pi,1} = Q_\pi$. Hence, such defined GVFs vector has progressive dependence. Appendix C provides further details and more examples in RL. More formally, we refer to this structure of forward GVF with progressive dependence as casual filtering as defined below. Note that a similar structure was called acyclic graph in [33].

**Definition 1** (Forward GVF with causal filtering). *For a given policy $\pi$, a forward GVF $G_\pi = [G_{\pi,1}^\top, \cdots, G_{\pi,k}^\top]^\top$ with causal filtering are associated with signals satisfying*

$$C_i = B_i + \sum_{j=1}^{i-1} A_{i,j} G_{\pi,j} \quad for \ 2 \le i \le k,$$

*where $C_i$ and $G_{\pi,j}$ are obtained by respectively stacking $C_i(s,a) \in \mathbb{R}^{d_i}$ and $G_{\pi,j}(s,a) \in \mathbb{R}^{d_i}$ over $\mathcal{S} \times \mathcal{A}$, $B_i \in \mathbb{R}^{d_i|\mathcal{S}||\mathcal{A}|}$ is an observable signal, and the coefficient matrix $A_{i,j} \in \mathbb{R}^{d_i|\mathcal{S}||\mathcal{A}| \times d_j|\mathcal{S}||\mathcal{A}|}$ captures how the $j$-th GVF $G_{\pi,j}$ affects the $i$-th accumulation signal $C_i$. Further, $B_i$ and $A_{i,j}$ are bounded to ensure $G_{\pi,i}$ to be well defined.*

Definition 1 indicates that all GVFs are interrelated with a causal filtering structure, i.e., each signal $C_i$ is a linear function of all lower-indexed $G_{\pi,l}$ for $1 \le l < i$. Definition 1 also implies that the forward GVF $G_\pi = [G_{\pi,1}^\top, \cdots, G_{\pi,k}^\top]^\top$ with causal filtering satisfies the following **lower-triangular Bellman equation** given by

$$G_\pi = \mathcal{T}_{G,\pi} G_\pi = B + M_\pi G_\pi, \tag{2}$$

where $\mathcal{T}_{G,\pi}$ denotes the forward general Bellman operator (GBO), $B = [B_1^\top, \cdots, B_k^\top]^\top$ and

$$M_\pi = \begin{bmatrix} \gamma_1 \bar{P}_{\pi,1} & 0 & \cdots & 0 \\ A_{2,1} & \gamma_2 \bar{P}_{\pi,2} & \cdots & 0 \\ \vdots & \vdots & & \vdots \\ A_{k,1} & A_{k,2} & \cdots & \gamma_k \bar{P}_{\pi,k} \end{bmatrix}.$$

where $\bar{P}_{\pi,i} = [\mathsf{P}_\pi \otimes \mathsf{I}_{d_i}]$. Clearly, the canonical value function $Q_\pi$ and Bellman operator $\mathcal{T}_\pi$ defined in eq. (1) is a special case of $G_\pi$ and $\mathcal{T}_{G,\pi}$ defined in eq. (2).

## 2.2  Backward General Value Function

In contrast to the forward GVF defined in the last section, which represents the *predictive knowledge*, in some RL scenarios, we also want to capture the *retrospective knowledge*, which represents the accumulation of signals that have been collected from the past. Consider a set of GVFs $\hat{G}_\pi = [\hat{G}_{\pi,1}^\top, \cdots, \hat{G}_{\pi,k}^\top]^\top$, where each GVF $\hat{G}_{\pi,i}$ is defined as the *backward* accumulation of a vector signal $\hat{C}_i(s,a) \in \mathbb{R}^{d_i}$ given by $\hat{G}_{\pi,i}(s,a) = \mathbb{E}\big[\sum_{t=-\infty}^{0} \gamma_i^{-t}\hat{C}_i(s_t,a_t)\big|(s_0,a_0) = (s,a),\pi\big]$. To distinguish from the forward GVF $G_{\pi,i}$ defined in Section 2.1, we denote $\hat{G}_{\pi,i}$ as the backward GVF. For general purpose, we also consider the causal filtering setting for $\hat{G}_\pi$, in which each $\hat{C}_i(s,a)$ depends on the lower-indexed value functions $\hat{G}_{\pi,1}, \cdots, \hat{G}_{\pi,i-1}$ in the set. We define the backward GVF with causal filtering as follows.

**Definition 2** (Backward GVF with causal filtering). *For a given policy $\pi$, a backward GVF $\hat{G}_\pi = [\hat{G}_{\pi,1}^\top, \cdots, \hat{G}_{\pi,k}^\top]$ with causal filtering are associated with signals satisfying*

$$\hat{C}_i = B_i + \sum_{j=1}^{i-1} A_{i,j} \hat{G}_{\pi,j} \quad for \ 2 \le i \le k,$$

*where $\hat{C}_i$ and $\hat{G}_{\pi,j}$ are obtained by respectively stacking $\hat{C}_i(s,a) \in \mathbb{R}^{d_i}$ and $\hat{G}_{\pi,j}(s,a) \in \mathbb{R}^{d_i}$ over $\mathcal{S} \times \mathcal{A}$, $B_i \in \mathbb{R}^{d_i|\mathcal{S}||\mathcal{A}|}$ is an observable signal, and the coefficient matrix $A_{i,j} \in \mathbb{R}^{d_i|\mathcal{S}||\mathcal{A}| \times d_j|\mathcal{S}||\mathcal{A}|}$ captures how the $j$-th GVF $\hat{G}_{\pi,j}$ affects the $i$-th accumulation signal $\hat{C}_i$. Further, $B_i$ and $A_{i,j}$ are bounded to ensure $\hat{G}_{\pi,i}$ to be well defined.*

For an ergodic MDP that starts from $-\infty$, we have $(s_{t-1}, a_{t-1}) \sim \mu_\pi(\cdot)$, $(s_t, a_t) \sim \mathsf{P}_\pi(\cdot|s_{t-1}, a_{t-1})$, and $(s_t, a_t) \sim \mu_\pi(\cdot)$ for all $-\infty < t < \infty$. The Bayes' theorem implies that

$$P((s_{t-1}, a_{t-1})|(s_t, a_t)) = \frac{\mu_\pi(s_{t-1}, a_{t-1})\mathsf{P}_\pi((s_t, a_t)|(s_{t-1}, a_{t-1}))}{\mu_\pi(s_t, a_t)}. \tag{3}$$

The reverse conditional probability in eq. (3) together with the definition of backward GVF in Definition 2 implies that the backward GVFs $\hat{G}_\pi = [\hat{G}_{\pi,1}^\top, \cdots, \hat{G}_{\pi,k}^\top]^\top$ with causal filtering satisfies

$$\hat{G}_\pi = \hat{\mathcal{T}}_{G,\pi}\hat{G}_\pi = B + \hat{M}_\pi \hat{G}_\pi, \tag{4}$$

where $\hat{\mathcal{T}}_{G,\pi}$ denotes the backward GBO, $B = \left[B_1^\top, \cdots, B_k^\top\right]^\top$, and

$$\hat{M}_\pi = \begin{bmatrix} \gamma_1 \hat{P}_{\pi,1} & 0 & \cdots & 0 \\ A_{2,1} & \gamma_2 \hat{P}_{\pi,2} & \cdots & 0 \\ \vdots & \vdots & & \vdots \\ A_{k,1} & A_{k,2} & \cdots & \gamma_k \hat{P}_{\pi,k} \end{bmatrix},$$

where $\hat{P}_{\pi,i} = U_{\pi,i}^{-1}[\mathsf{P}_\pi \otimes \mathsf{I}_{d_i}]U_{\pi,i}$.

## 3 Off-Policy Evaluation of GVFs

### 3.1 Problem Formulation

In this paper, we study the GVFs evaluation problem for a target policy $\pi$. We focus on the *behavior-agnostic off-policy* setting, in which we have access only to samples generated from an off-policy (i.e., a behavior policy) with the distribution $\mathcal{D}$, i.e., $(s_j, a_j, B_j, s_j') \sim \mathcal{D}$ $(j > 0)$. Specifically, the state-action pair $(s_j, a_j)$ is sampled from a possibly **unknown** distribution $D(\cdot) : \mathcal{S} \times \mathcal{A} \to [0,1]$, $B_j = [B_1(s_j, a_j), \cdots, B_k(s_j, a_j)]$ is an observable signal vector, and the successor state $s_i'$ is sampled from $\mathsf{P}(\cdot|s_i, a_i)$. Our goal is to design an efficient algorithm to estimate $G_\pi$ (or $\hat{G}_\pi$) given the sample set $\{(s_j, a_j, B_j, s_j')\}_{j>0}$. We make the following dataset coverage assumption.

**Assumption 1.** *We assume that $D(s,a) > 0$ for all $(s,a) \in \mathcal{S} \times \mathcal{A}$.*

### 3.2 Linear Function Approximation

When $|\mathcal{S}|$ is large, a linear function can be used to approximate the GVF: $G_{\pi,i}(s,a) \approx G_{\pi,i}(\theta_i; s, a) = \theta_i^\top \phi_i(s,a) = [\phi_i(s,a)^\top \otimes \mathsf{I}_{d_i}]\mathrm{vec}(\theta_i^\top)$, where $\phi_i(s,a) \in \mathbb{R}^{K_i}$ is the feature vector, and $\theta_i \in \mathbb{R}^{K_i \times d_i}$ is a learnable weight matrix. In the sequel, we omit $\pi$ in $G_{\pi,i}$ and use the notation $G_i$. We make the following assumption for linear feature $\phi$, which is standard in linear function approximation setting [45, 23, 55].

**Assumption 2.** *We assume that $\|\phi_i(s,a)\|_2 \le 1$ for all $i = 1, \cdots, k$ and $(s,a) \in \mathcal{S} \times \mathcal{A}$.*

The linear approximation can then be written as $G_i(\theta_i) = [\Phi_i \otimes \mathsf{I}_{d_i}]\mathrm{vec}(\theta_i^\top)$, where $\Phi_i$ is the base matrix obtained by stacking $\phi_i(s,a)^\top$ over $\mathcal{S} \times \mathcal{A}$. To ensure the uniqueness of the solution $\theta_i$, we assume that $\Phi_i$ has linearly independent columns. The joint vector of GVFs can be denoted as $[G_1^\top(\theta_1), \cdots, G_k^\top(\theta_k)]^\top$, which is captured by the joint parameters $\theta = [\mathrm{vec}(\theta_1^\top)^\top, \cdots, \mathrm{vec}(\theta_k^\top)^\top]^\top \in \mathbb{R}^{\sum_{i=1}^k K_i d_i}$. Then the function approximation of GVFs can be written more compactly as $G(\theta) = \Phi\theta$, where $\Phi = \mathrm{diag}([\Phi_1 \otimes \mathsf{I}_{d_1}], \cdots, [\Phi_k \otimes \mathsf{I}_{d_k}])$. For each $(s,a)$, the linear function approximation associated with each $(s,a)$ can be written as $G(\theta; s, a) = \phi(s,a)\theta$, where $\phi(s,a) = \mathrm{diag}([\phi_1(s,a)^\top \otimes \mathsf{I}_{d_1}], \cdots, [\phi_k(s,a)^\top \otimes \mathsf{I}_{d_k}])$. We define the linear function space spanned by the columns of the feature matrix $\Phi$ as $\mathcal{F}_\Phi = \{\Phi\theta | \theta \in R_\theta\}$, in which $R_\theta$ is a convex set. Given the function class $\mathcal{F}_\Phi$, the evaluation problem of GVFs amounts to searching for a parameter $\theta^* \in R_\theta$ such that $G(\theta^*)$ approximates $G_\pi$ (or $\hat{G}_\pi$) well. In the sequel, we use $\bar{\mathcal{T}}_{G,\pi}$ to represent $\mathcal{T}_{G,\pi}$ or $\hat{\mathcal{T}}_{G,\pi}$, interchangeably, based on the context.

### 3.3 A New Off-policy GVF Evaluation Approach

**Drawbacks of GTD.** In previous works, the gradient TD (GTD) method [45, 23] has been used for policy evaluation (including GVF evaluation) in the off-policy setting [37, 69, 70, 61]. GTD adopts the Mean Squared Projected Bellman Error (MSPBE) for GVF evaluation with linear function approximation, which is given by

$$\hat{\theta}^* = \mathrm{argmin}_{\theta \in R_\theta} \, \mathrm{MSPBE}(\theta) \triangleq \mathbb{E}_D \left[ \left\| G(\theta; s, a) - \Gamma_{\mathcal{F}_\Phi, D} \bar{\mathcal{T}}_{G, \pi} G(\theta; s, a) \right\|_2^2 \right]. \tag{5}$$

where $\Gamma_{\mathcal{F}_\Phi, D}$ denotes the projection operator onto the space $\mathcal{F}_\Phi$ w.r.t. the $\|\cdot\|_D$–norm, i.e., for any vector function $f(s, a)$ of $(s, a)$, we have $\Gamma_{\mathcal{F}_\Phi, d} f = G(\theta_f)$, in which $\theta_f = \mathrm{argmin}_{\theta \in R_\theta} \|f - G(\theta)\|_D$. One drawback of GTD is that the expectation in the objective function is taken over the off-policy sampling distribution $D(\cdot)$, which does not exactly reflect the desirable evaluation under the **target** policy. As the result, the optimal point of GTD ($\hat{\theta}^*$) can still have a large approximation error with respect to the **ground truth** value of GVF, even if the approximation function class is arbitrarily expressive. More detailed discussion about GTD is provided in Appendix D.

**Generalized Temporal Difference (GenTD) Learning.** In this work, we propose a novel unified approach to evaluate both the forward and backward GVFs in the off-policy setting, which we refer to as generalized temporal difference (GenTD) learning. Specifically, we aim to learn $\theta^*$ for GVF evaluation by minimizing the mean-squared projected general Bellman error (MSPGBE) defined as

$$\theta^* = \mathrm{argmin}_{\theta \in R_\theta} \, \mathrm{MSPGBE}(\theta) \triangleq \mathbb{E}_{\mu_\pi} \left[ \left\| G(\theta; s, a) - \Gamma_{\mathcal{F}_\Phi, \mu_\pi} \bar{\mathcal{T}}_{G, \pi} G(\theta; s, a) \right\|_2^2 \right], \tag{6}$$

where recall that $\bar{\mathcal{T}}_{G, \pi}$ represents the GBO of either forward or backward GVFs. In contrast to GTD, the objective function in eq. (6) takes the expectation over the stationary distribution $\mu_\pi$ of the target distribution, which precisely captures the desired goal of GVF evaluation under the target policy. On the other hand, such an objective does cause implementation challenge, because the data samples are generated by the behavior policy, so that estimators based on such data directly can incur a large bias error. To solve such an issue, we will apply the density ratio $\rho(s, a) = \mu_\pi(s, a)/D(s, a)$ to adjust the distribution and further adopt the GenDICE/GradientDICE method proposed in [65, 67] to estimate $\rho(s, a)$ during the execution of the algorithm.

To describe our algorithm GenTD (see Algorithm 1), we first note that eq. (6) implies the following optimality condition for $\theta^*$ and all $f \in \mathcal{F}_\Phi$,

$$\langle G(\theta^*; \cdot) - \bar{\mathcal{T}}_{G; \pi} G(\theta^*; \cdot), f(\cdot) - G(\theta^*;) \rangle_{\mu_\pi} \geq 0,$$

or equivalently

$$\langle g(\theta^*), \theta - \theta^* \rangle \geq 0, \quad \forall \theta \in R_\theta, \tag{7}$$

where $g(\theta) = \Phi^\top U_\pi (G(\theta) - \bar{\mathcal{T}}_{G, \pi} G(\theta))$. The variational inequality theory in (Chapter 3 [20]) suggests that under an appropriately chosen stepsize $\alpha_t$, the update $\theta_{t+1} = \Gamma_{R_\theta}(\theta_t - \alpha_t g(\theta_t))$ converges to the optimal point $\theta^*$, where $\Gamma_{R_\theta}$ denotes the projection operator onto the set $R_\theta$ in terms of the Euclidean norm. However, since it is intractable to explicitly compute $g(\theta)$ in practice, we usually estimate $g(\theta)$ using random samples. In the off-policy setting, consider a sample $x = (s, a, s', a')$, in which $(s, a) \sim D(\cdot)$, $s' \sim \mathsf{P}(\cdot|s, a)$, and $a' \sim \pi(\cdot|s')$, we can formulate the following update rule:

$$\theta_{t+1} = \theta_t - \alpha_t \hat{\rho}(s, a) g(x, \theta_t), \tag{8}$$

where $\hat{\rho}(s, a)$ is an approximation of the density ratio $\rho(s, a) = \mu_\pi(s, a)/D(s, a)$, $g(x, \theta) = -\phi(s, a)^\top \delta(x, \theta)$ for forward GVFs and $g(x, \theta) = -\phi(s', a')^\top \delta(x, \theta)$ for backward GVFs, where $\delta(x, \theta)$ is the temporal difference error defined as $\delta(x, \theta) = B(s, a) + m(x)\phi(s', a')\theta - \phi(s, a)\theta$ for forward GVFs, and $\delta(x, \theta) = B(s', a') + \hat{m}(x)\phi(s, a)\theta - \phi(s', a')\theta$ for backward GVFs. Here $m$ and $\hat{m}$ are matrices that capture the correlations between difference estimations in forward and backward GVFs evaluation settings, respectively. Here we adopt the GenDICE/GradientDICE method that proposed in [65, 67] to learn $\rho(s, a)$. In previous works, GenDIC/GradientDICE has only been used for estimating the scalar value $J_\pi = \mathbb{E}_{\mu_\pi}[r(s, a)]$ in the off-policy setting [65, 70, 53]. Our work is the first to adapt this method to solve the more challenging off-policy GVFs evaluation problem.

---

**Algorithm 1** Generalized TD Learning (GenTD)

---

**Initialize:** Approximator parameters $w_{f,0}$, $w_{\rho,0}$ and $\theta_0$
**for** $t = 0, \cdots, T-1$ **do**
  Obtain sample $(s_t, a_t, C_t, s_t') \sim \mathcal{D}_d$ and $a_t' \sim \pi(\cdot|s_t')$
  $\bar{\delta}_t = \psi_t^\top \theta_{\rho,t}(\psi_t' - \psi_t)$
  $\eta_{t+1} = w_{\rho,t} + \beta_t(\psi_t^\top w_{\rho,t} - 1 - \eta_t)$
  $w_{f,t+1} = w_{f,t} + \beta_t(\bar{\delta}_t - \psi_t^\top w_{f,t}\psi_t)$
  $w_{\rho,t+1} = \Gamma_{R_\rho}\left(w_{\rho,t} - \beta_t(\psi_t'^\top w_{f,t}\psi_t - \psi_t^\top w_{f,t}\psi_t + \eta_t\psi_t)\right)$
  $\theta_{t+1} = \Gamma_{R_\theta}\left(\theta_t - \alpha_t[w_{\rho,t}^\top \psi(s_t, a_t)]g(x_t, \theta_t)\right)$
  **Forward GVF:** $g(x, \theta) = -\phi(s,a)^\top (B(s,a) + m(x)\phi(s',a')\theta - \phi(s,a)\theta)$
  **Backward GVF:** $g(x, \theta) = -\phi(s',a')^\top (B(s',a') + \hat{m}(x)\phi(s,a)\theta - \phi(s',a')\theta)$
**end for**

---

**Learning Density Ratio.** GenDICE/GradientDICE estimates the density ratio $\rho(s, a)$ via solving the following min-max problem [65, 67]:

$$\min_{\rho} \max_{f,\eta} L(\hat{\rho}, f, \eta) := \mathbb{E}_{\mathcal{D}}[\hat{\rho}(f' - f)] - \frac{1}{2}\mathbb{E}_{\mathcal{D}}[f^2] + \mathbb{E}_{\mathcal{D}}[\eta\hat{\rho} - \eta] - \frac{1}{2}\eta^2. \tag{9}$$

We parameterize both $\rho$ and $f$ by linear function with linearly independent features $\psi \in \mathbb{R}^{d_\rho}$, i.e., $\hat{\rho}(s, a; w_\rho) = \psi(s, a)^\top w_\rho$ and $\hat{f}(s, a; w_f) = \psi(s, a)^\top w_f$ for all $(s, a) \in \mathcal{S} \times \mathcal{A}$. To guarantees the stability of the density ratio learning, we assume that the matrix $A = \mathbb{E}_{\mathcal{D}\cdot\pi}[\psi(\psi - \psi')^\top]$ is non-singular. Note that this assumption can be removed by adding an $l_2$–regularizer in eq. (9). In GenTD (see Algorithm 1), we estimate the density ratio via updating the parameter $w_{\rho,t}$ iteratively. The density estimator $\hat{\rho}(s_t, a_t; w_{\rho_t}) = \psi(s_t, a_t)^\top w_{\rho_t}$ is then used to reweight the update $g(x_t, \theta_t)$.

**Comparison between GenTD and GTD.** Compared with GTD, our GenTD has the following two advantages. First, since GTD does not adjust the distribution mismatch of sampling, the optimal point of GTD can suffer from large approximation error with respect to the ground truth GVFs even with highly expressive function classes. In contrast, the optimum of GenTD is guaranteed to approximate the ground truth GVFs well with sufficiently expressive function classes. Second, GTD needs to update a high-dimensional auxiliary parameter $w$ simultaneously with $\theta$ to stabilize the convergence, where $w \in \mathbb{R}^{\sum_{i=1}^k K_i d_i}$ has the same dimension as $\theta \in \mathbb{R}^{\sum_{i=1}^k K_i d_i}$ (note that $\sum_{i=1}^k K_i d_i$ can be very large in the high dimensional regime or when the number of GVFs $k$ is very large). Such an update of $w$ can be very costly. In contrast, GenTD introduces only low-dimensional auxiliary parameters $[w_\rho, w_f, \eta] \in \mathbb{R}^{2d_\rho+1}$ for density ratio estimation, which is more efficient than GTD since $d_\rho$ could be much smaller than $\sum_{i=1}^k K_i d_i$.

## 4 Main Theorems

In this section, we develop the finite-time convergence rate for our Off-GenTD algorithm. To this end, we first want to establish a certain contraction property for the general Bellman operator of interest here. Although the contraction property has been proven in the canonical value function settings [55, 70], it is unclear whether such a property still holds for **multiple multi-dimensional and interrelated** GVFs. We will next establish that such a property still holds for both forward and backward GVFs with causal filtering, but needs to be under a properly chosen norm.

Consider the GVFs vector $G_\pi = [G_{\pi,1}^\top, \cdots, G_{\pi,k}^\top]^\top$. We define a norm $\|\cdot\|_{\mu_\pi,\alpha}$ associated with a weighting vector $\alpha = [\alpha_1, \cdots, \alpha_k] \in \Delta_k$, where $\Delta_k$ denotes the simplex in $k$-dimensional space, as $\|G_\pi\|_{\mu_\pi,\alpha} = \sum_{i=1}^k \alpha_i \|G_{\pi,i}\|_{\mu_\pi}$, where $0 < \alpha_i \leq 1$ for all $i$ and $\sum_{i=1}^k \alpha_i = 1$. We also define $\gamma_{\max} := \max_{i=1\cdots,k} \gamma_i$, which is strictly less than 1.

**Proposition 1** (Contraction of Forward/Backward GBO). [1] *For any $G_\pi, G_\pi' \in \mathbb{R}^{|\mathcal{S}||\mathcal{A}|\sum_{i=1}^k K_i d_i}$, there exists a weighting vector $\alpha$ such that*

$$\left\|\overline{\mathcal{T}}_{G,\pi}G_\pi - \overline{\mathcal{T}}_{G,\pi}G_\pi'\right\|_{\mu_\pi,\alpha} \leq \frac{1+\gamma_{\max}}{2}\|G_\pi - G_\pi'\|_{\mu_\pi,\alpha}, \tag{10}$$

---

[1]The contraction property of GBO for *forward* GVFs has been proved in [33] but under different assumptions and with respect to a different norm. The result for *backward* GVFs is first established in our work.

where $\overline{\mathcal{T}}_{G,\pi}$ can be either $\mathcal{T}_{G,\pi}$ (forward GBO, eq. (2)) or $\hat{\mathcal{T}}_{G,\pi}$ (backward GBO, eq. (4)).

Despite the correlations between GVFs, Proposition 1 shows that the contraction property is still preserved under a properly chosen norm for $\mathcal{T}_{G,\pi}$ and $\hat{\mathcal{T}}_{G,\pi}$ in forward and backward GVF settings, respectively. The norm can vary for different GVFs. Proposition 1 also implies that both forward and backward GVFs ($G_\pi$ and $\hat{G}_\pi$) can be identified as unique fixed point of their corresponding GBOs.

Based on Proposition 1, we next establish the monotonicity property for our GenTD algorithm, if it takes the population update $g(\theta) = \Phi^\top U_\pi(G(\theta) - \overline{\mathcal{T}}_{G,\pi}G(\theta))$.

**Proposition 2** (Monotonicity). *Suppose Assumption 1 &2 hold. Consider the globally optimal point $\theta^*$ defined in eq. (7). There exists a constant $\lambda_G$ such that for all $\theta \in R_\theta$, we have*

$$\langle g(\theta^*) - g(\theta), \theta^* - \theta \rangle \geq \lambda_G \|\theta - \theta^*\|_F^2, \tag{11}$$

*where $\lambda_G := (1 - \gamma_{\max}) \min_{1 \leq i \leq k} \zeta_i$ and $\zeta_i := \lambda_{\min}(\Phi_i^\top U_\pi \Phi_i)$.*

Proposition 2 implies the contraction property of $g(\theta)$. It guarantees that $\theta$ moves towards a globally optimal point $\theta^*$ if it is updated along the direction $-g(\theta)$. Proposition 2 generalizes the monotonicity property to a much broader class of interrelated and multi-dimensional GVF evaluation, which is far more beyond TD learning for the value function evaluation studied in [55, 70]. The following theorem characterizes the convergence rate of GenTD.

**Theorem 1.** *Suppose Assumption 1 &2 hold. Consider the GenTD update in Algorithm 1. Let the stepsize $\alpha_t = \Theta(t^{-1})$ and $\beta_t = \Theta(t^{-1})$. We have*

$$\mathbb{E}[\|\theta_T - \theta^*\|_F^2] \leq \mathcal{O}\left(\frac{\|\theta_0 - \theta^*\|_F^2}{T^2}\right) + \mathcal{O}\left(\frac{1}{\lambda_G^3 T}\right) + \mathcal{O}\left(\frac{\varepsilon_\rho}{\lambda_G^2}\right), \tag{12}$$

*where $\varepsilon_\rho = \sqrt{\mathbb{E}_{\mathcal{D}\cdot\pi}[\hat{\rho}(s, a; w_\rho^*) - \rho(s, a)]^2}$ is the approximation error introduced by the density ratio learning, with $w_\rho^*$ defined in eq. (9).*

Theorem 1 shows that GenTD converges to the globally optimal point $\theta^*$ at a rate $\mathcal{O}(1/T)$. The convergence speed of $\theta$ also depends on the conditional number $\lambda_G$, where the converge becomes faster as $\lambda_G$ increases. Specifically, the R.H.S. of eq. (12) consists of three terms. The first term corresponds to the initialization error, which delays as fast as $\mathcal{O}(1/T^2)$. The second term corresponds to the variance error, which dominates the convergence rate of GenTD to be $\mathcal{O}(1/T)$. The last term corresponds to a non-vanishing optimality gap, which is introduced by the function approximation error in the density ratio estimation, and decreases as the expressive power of the approximation function class $\{\hat{\rho}(w_\rho) : w_\rho \in R_\rho\}$ increases. For more discussion about this approximation error, please refer to [65, 67]. The convergence analysis of GenTD is more challenging than that of TD learning [2, 7, 40] and GTD [62, 17], as we need to handle an additional approximation error introduced by the *dynamically changing* density ratio estimator $\hat{\rho}(w_{\rho_t})$.

Theorem 1 establishes the convergence of GenTD to the globally optimal point $\theta^*$ of the objective function in eq. (6), which provides the value estimation $G(\theta^*)$ for the GVFs. We are then interested in characterizing how close such an estimation is to the ground truth GVF $G_\pi$, which is our ultimate goal of evaluation. We characterize this in the following theorem.

**Theorem 2** (Convergence of GenTD to Ground Truth). *Consider $\theta^*$ defined in eq. (6). Suppose the same conditions in Proposition 1 & 2 hold. We have*

$$\|G(\theta^*) - G_\pi\|_{\mu_\pi,\alpha} \leq \frac{1}{1-\gamma_G} \|\Gamma_{\mathcal{F}_\Phi,\mu_\pi} G_\pi - G_\pi\|_{\mu_\pi,\alpha}.$$

Theorem 2 indicates that the distance between the optimal estimation $G(\theta^*)$ and the true GVF $G_\pi$ is upper bounded by the approximation error of the function class $\mathcal{F}_\Phi$ for the ground truth GVF $G_\pi$ (note that $\Gamma_{\mathcal{F}_\Phi,\mu_\pi} G_\pi$ denotes the projection of $G_\pi$ to the function approximation class $\mathcal{F}_\Phi$). Hence, Theorem 2 guarantees that $G(\theta^*)$ can be as close as possible to the true GVF $G_\pi$, as long as the function class $\mathcal{F}_\Phi$ is sufficiently expressive. In particular, if $\mathcal{F}_\Phi$ is complete, i.e., there exists $G_\theta \in \mathcal{F}_\Phi$ such that $G_\theta = G_\pi$, then GenTD is guaranteed to converge exactly to the ground truth $G_\pi$. Note that Theorem 2 is the first result of such a type developed for both forward and backward GVFs.

**Comparison between GenTD and GTD.** If $\mathcal{F}_\Phi$ is complete, GTD performs similarly to GenTD and is guaranteed to converge to the ground truth $G_\pi$ (see Appendix D.2 for the proof). The major difference between GenTD and GTD occurs when $\mathcal{F}_\Phi$ is not complete. In such a case, our GenTD

still maintains the desirable performance as guaranteed by Theorem 2, but the optimal point of GTD (i.e., $\hat{\theta}^*$ in eq. (5)) does not have guaranteed convergence to the ground truth. As shown in [19, 10, 27], even in the value function evaluation setting (a special case of forward GVF evaluation) the approximation error $\|G(\hat{\theta}^*) - G_\pi\|_D$ of GTD can be arbitrarily poor even if $\mathcal{F}_\Phi$ can represent the true value function arbitrarily well (but not exactly). Such a disadvantage of GTD is mainly due to the distribution mismatch in its objective function as we discuss in Section 3.3.

In the backward GVFs evaluation setting, GTD can perform even worse. As we show in the following example, GTD may fail to learn the ground truth $G_\pi$ even if the function class $\mathcal{F}_\Phi$ is complete. Note that for such a case, GenTD converges to the ground truth as guaranteed by Theorem 2.

**Example 1** (GTD Fails for Complete $\mathcal{F}_\Phi$). *Consider a three-state Markov chain, with transition kernel $\mathsf{P} = [[0.1, 0.9, 0], [0.1, 0, 0.9], [0, 0.1, 0.9]]^\top$, discount factor $\gamma = 0.99$, and the reward function $R = [1, 0, 1]^\top$. The back value function in this MDP is given by $\bar{V} = [8.1555, 9.0389, 9.0184]^\top$. Suppose GTD is applied to solving the evaluation problem with the parameter space $R_\theta = \mathbb{R}$. Then, there exists an off-policy distribution $D$ such that using the perfect bases $\Phi = [8.1555, 9.0389, 9.0184]^\top$, the optimal point $\bar{\theta}^*$ learned by GTD still has non-zero approximation error, i.e., $\|\Phi\bar{\theta}^* - \bar{V}\|_D \geq 3$.*

## 5 Experiments

We conduct empirical experiments to answer the following two questions: (a) can GenTD evaluate both the forward and backward GVFs efficiently? (2) how does GenTD compare with GTD in terms of the convergence speed and the quality of the estimation results? In our experiments, we

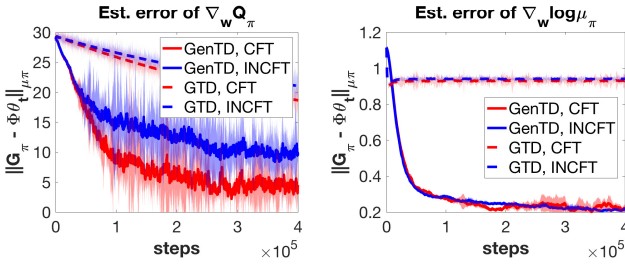

Figure 1: *Comparison between GenTD and GTD for the tasks of evaluating $\nabla_w Q_\pi$ and $\nabla_w \log \mu_\pi$.*

consider a variant of Baird's counterexample [1, 44] with 7 states and 2 actions (see Figure 2 in Appendix A). We study the problem of evaluating two high-dimensional GVFs, the gradient of Q-function: $\nabla_w Q_\pi \in \mathbb{R}^{14}$ (forward GVF), and the gradient of logarithmic stationary distribution: $\nabla_w \log(\mu_\pi) \in \mathbb{R}^{14}$ (backward GVF), associated with a soft-max policy parameterized by $w \in \mathbb{R}^{14}$. We consider two types of feature matrices $\Phi$ for estimating the GVFs: complete feature (CFT) and incomplete feature (INCFT), where CFT has large enough expressive power so that the ground true GVF can be fully expressed by the function class $\mathcal{F}_\Phi$, whereas INCFT does not have enough expressive power and cannot capture the ground true GVF exactly. The discount factor $\gamma$ is set to be 0.99 in all tasks, and all curves in the plots are averaged over 20 independent runs. The detailed experimental setting is provided in Appendix A.

The learning curves for GenTD and GTD are provided in Figure 1. We evaluate their performances based on the estimation error with respect to the ground truth GVF: $\|\Phi\theta_t - G_\pi\|_{\mu_\pi}$. Note that both $\nabla_w Q_\pi$ and $\nabla_w \log \mu_\pi$ can be exactly computed in tabular setting, so that the estimator error of the ground truth can be computed. For the task of $\nabla_w Q$ evaluation, GenTD converges considerably faster and closer to the ground truth (i.e., smaller estimation error) than GTD. which can be attributed to the larger conditional number $\lambda_G$ of GenTD. For the task of $\nabla_w \log \mu_\pi$ evaluation, GenTD moves fast towards the ground truth GVF, whereas GTD, although still converges, stays far away from the ground truth GVF even with CFT, which matches with our Example 1. As we discuss in Section 4, this is because GTD in the backward GVF evaluation setting has distribution mismatch in its objective function, which can significantly shift the optimal point from the ground truth GVF.

# 6  Conclusion

We studied the off-policy evaluation problem of both forward and backward GVFs. We focused on the class of GVFs with casual filtering, which covers a wide range of multiple interrelated and possibly high-dimensional GVFs. We first showed that GVFs in such a class is the fixed point of a general Bellman operator. Based on such a property, we proposed a new off-policy algorithm called GenTD. GenTD evaluates GVFs efficiently by jointly updating the GVF approximation parameter and a density ratio estimator, which adjusts the mismatch of the behavior policy and assists the convergence to the ground truth GVFs. We show that GenTD provably converges to the globally optimal point, and such an optimal point is guaranteed to converge to the ground truth GVFs as long as the function expressive power is sufficiently large. For future work, it is interesting to study nonlinear function approximation for GVFs evaluation.

# 7  Acknowledge

The work of T. Xu and Y. Liang was supported in part by the U.S. National Science Foundation under the grant 2148253 and 1761506.

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
