# Supplementary Materials

## A    Specification of Experiments

The Baird's counterexample [1, 44] is shown in Figure 2. There are two actions represented by solid line and dash line, respectively. The the *dash* action leads to states 1-6 with equal probability and a reward $+1$, and *solid* action always leads to state 7 and a reward 0. The behavior distribution over the state-action space $(s, a)$ is given as

$$D(\cdot) = \begin{cases} D(s_1, a_1) = 0.2, & D(s_1, a_2) = 0.1, \\ D(s_2, a_1) = 0.2, & D(s_2, a_2) = 0.1, \\ D(s_3, a_1) = 0.04, & D(s_3, a_2) = 0.04, \\ D(s_4, a_1) = 0.04, & D(s_4, a_2) = 0.04, \\ D(s_5, a_1) = 0.04, & D(s_5, a_2) = 0.04, \\ D(s_6, a_1) = 0.04, & D(s_6, a_2) = 0.04, \\ D(s_7, a_1) = 0.04, & D(s_7, a_2) = 0.04, \end{cases}$$

where $s_i$ denotes state "$i$" ($i = 1, \cdots, 7$), $a_1$ denotes the *dash* action and $a_2$ denotes the *solid* action.

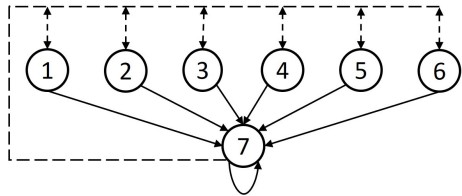

Figure 2: A variant of Baird's counterexample.

We consider the soft-max policy given as

$$\pi_w(s_i, a_j) = \frac{\exp(w_{2(i-1)+j})}{\exp(w_{2(i-1)+1}) + \exp(w_{2(i-1)+2})}, \quad i = \{1, \cdots, 7\}, \ j = \{1, 2\},$$

where $w \in \mathbb{R}^{14}$ is the parameter of the policy given as

$$w^\top = [0.0, 1.8, 0.0, 1.8, 0.0, 1.8, 0.0, 1.8, 0.0, 1.8, 0.0, 1.8, 0.0, 1.8].$$

The complete feature (CFT), incomplete feature (INCFT) and the learning rate for each task are given as follows:

- **Evaluation of $\nabla_w Q_\pi$ (forward GVF).** In this task, we need to evaluate both $Q_\pi$ and $\nabla_w Q_\pi$ (see Appendix C.1 for detailed discussion about correlation between $Q_\pi$ and $\nabla_w Q_\pi$). Let the complete feature matrix $\Phi$ be the identity matrix, i.e., $\Phi = I \in \mathbb{R}^{14 \times 14}$. We let CFT for each $(s, a)$ be one of the rows of $\Phi$. We further remove one column of $\Phi$ to obtain the incomplete feature matrix $\Phi' \in \mathbb{R}^{14 \times 13}$. We let INCFT for each $(s, a)$ be one of the rows of $\Phi'$. For GenTD, the learning rate for updating $w_\rho$ and $\theta$ are 0.01 and 0.005, respectively. We use the same CFT and INCFT for the density ratio estimation as those for the $\nabla_w Q_\pi$ estimation. For GTD, the learning rate for both the main parameter $\theta$ and the auxiliary parameter $w$ are 0.005.

- **Evaluation of $\nabla_w \log \mu_\pi$ (backward GVF).** Appendix C.2 has detailed discussion about such a GVF. Note that this task corresponds to the setting where $\gamma = 1$. As discussed in Appendix H, the ground true GVF in this setting is in the space perpendicular to the vector $e = [1, \cdots, 1]^\top \in \mathbb{R}^{14}$. Here we use singular value decomposition (SVD) to obtain the complete feature matrix $\Phi \in \mathbb{R}^{14 \times 13}$ such that $\Phi\theta \neq Ce$ for all $\theta \neq 0$ and $C \neq 0$. We let CFT for each $(s, a)$ be one of the rows of $\Phi$. We further remove one column of $\Phi$ to obtain the incomplete feature matrix $\Phi' \in \mathbb{R}^{14 \times 12}$. We let INCFT for each $(s, a)$ be one of the rows of $\Phi'$. For GenTD, the learning rate for updating $w_\rho$ and $\theta$ are 0.05 and 0.005, respectively. We use the same CFT and INCFT for the density ratio estimation as those used in the above $\nabla_w Q_\pi$ estimation task. For GTD, the learning rate for both the main parameter $\theta$ and the auxiliary parameter $w$ are 0.005.

### A.1 Additional Experiments

In this subsection, we provide additional experiments in the task of evaluating $\nabla_w Q_\pi$ to explore the following two issues: (a) we demonstrate how GenTD performs with diminishing stepsize; and (b) we demonstrate how the bias error changes as the expressive power of the approximation function class changes. To this end, we remove one column of $\Phi$ with small weight and one column of $\Phi$ with large weight to obtain two incomplete feature matrices $\Phi_1'$ and $\Phi_2'$, respectively. We then let $\text{INCFT}_1$ and $\text{INCFT}_2$ for each $(s, a)$ be one of the rows of $\Phi_1'$ and $\Phi_2'$, respectively. Here, the linear function class with base $\text{INCFT}_1$ has larger expressive power than that with $\text{INCFT}_2$. In the evaluation of $\nabla_w Q_\pi$, we let the learning rate for updating $w_\rho$ and $\theta$ be $\alpha_t = \frac{1}{50+t}$ and $\beta_t = \frac{1}{100+t}$, respectively. The experiment result is given in Figure 3.

In Figure 3, we first observe that GenTD is able to achieve near zero bias error with CFT under diminishing stepsize. Second, the bias error of GenTD increases only slightly from CFT to $\text{INCFT}_1$, where the expressive power of $\text{INCFT}_1$ is still large. The bias error increases further under $\text{INCFT}_2$, which has lower expressive power. Overall, the bias error increases as the expressive power of the function approximation class decreases. Both observations are consistent with Theorem 3.

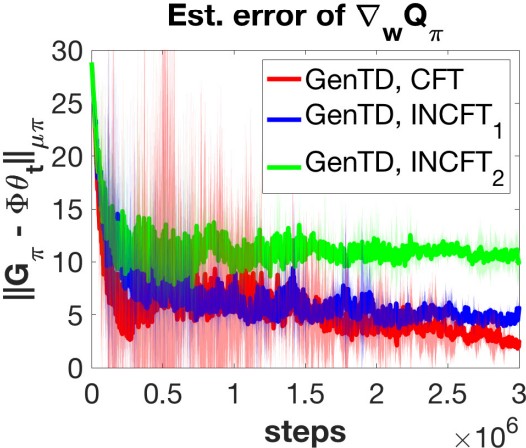

Figure 3: *Performance of GenTD in the task of evaluating $\nabla_w Q_\pi$ under diminishing stepsize.*

## B  Additional Related Works

The goal of OPE is to estimate the expected return of start states drawn randomly from a distribution. Importance sampling (IS) has been used for OPE in which sample rewards are reweighed to get unbiased value estimate of a new policy [29]. Later, doubly robust technique was proposed to reduce the variance of IS [16, 54, 22]. In the behavior policy agnostic setting, [28] proposed the GenDICE algorithm to estimate the IS with function approximation when performing the OPE, which also suffers less from the variance. Our approach GenTD is along the line of GenDICE in [28], which also adopts function approximation to estimate the density ratio. However, in our work we consider a more challenging setting in which we need to evaluate all the GVFs for each state-action pair instead of the mean of scalar value functions considered in [28].

## C  Examples of Forward and Backward GVFs

In this section, we present a number of example forward and backward GVF in RL applications.

### C.1  Examples of Forward GVFs

The forward GVF in Definition 1 arises naturally in the following RL applications.

**Case I: Variance of Reward-To-Go.**    In risk-sensitive domains such as finance, process control and clinical decision making [34, 25, 52, 50, 30, 31, 39, 15], in addition to the mean $J_\pi$

of the "reward-to-go", we are also interested in the variance of $J_\pi$ [34, 36], which is given by $\mathrm{Var}[J_\pi|(s_0,a_0)=(s,a)]=H_\pi(s,a)-Q_\pi^2(s,a)$, where $H_\pi$ is the second moment of $J_\pi$, i.e., $H_\pi(s,a)=\mathbb{E}[J_\pi^2|(s_0,a_0)=(s,a),\pi]$. [51] shows that $H_\pi$ satisfies

$$H_\pi = R^2 + 2\gamma M_R \mathsf{P}_\pi Q_\pi + \gamma^2 \mathsf{P}_\pi H_\pi, \tag{13}$$

where $R$ is defined in Section 2, $M_R = \mathrm{diag}(R) \in \mathbb{R}^{|\mathcal{S}||\mathcal{A}|\times|\mathcal{S}||\mathcal{A}|}$. Equation (13) implies that $H_\pi$ is the mean of the accumulation of signal $C(s,a) = r(s,a)^2 + 2\gamma\mathbb{E}[Q_\pi(s',a')|s,a]$ with discounted factor $\gamma^2$. Since $C(s,a)$ is a function of the reward $r(s,a)$ and value function $Q_\pi(s,a)$, we consider the joint vector of $Q_\pi$ and $H_\pi$ as $G_\pi$, i.e., $G_\pi = [Q_\pi^\top, H_\pi^\top]^\top$. We have that $G_\pi$ satisfies the general Bellman equation in eq. (2) with $B$ and $M_\pi$ specified as

$$B = \left[\begin{array}{c} R \\ R^2 \end{array}\right], \quad M_\pi = \left[\begin{array}{cc} \gamma\mathsf{P}_\pi & 0 \\ 2\gamma M_R\mathsf{P}_\pi & \gamma^2\mathsf{P}_\pi \end{array}\right]. \tag{14}$$

We consider the setting in which reward is bounded, i.e., $r(s,a) \le C_R$ for all $(s,a) \in \mathcal{S}\times\mathcal{A}$.

**Case II: Gradient of Q-function.** Suppose that the policy is parametrized by a smooth function $\pi_w$, in which $w \in \mathbb{R}^{d_w}$ is the parameter. Then the gradient $\nabla_w Q_\pi(s,a)$ of the Q-function w.r.t $w$ plays an important role in several RL applications such as variance reduced policy gradient [14] and on- and off-policy policy optimization [61, 38, 18, 5]. Specifically, [61, 18, 5] show that $\nabla_w Q_\pi$ satisfies:

$$\nabla_w Q_\pi = \gamma[\mathsf{P}_\pi \otimes \mathsf{I}_{d_w}][\nabla_w \Pi_{\pi_w} \cdot Q_\pi] + \gamma[\mathsf{P}_\pi \otimes \mathsf{I}_{d_w}]\nabla_w Q_\pi, \tag{15}$$

where $\nabla_w \Pi_{\pi_w} \in \mathbb{R}^{d_w|\mathcal{S}||\mathcal{A}|}$ is obtained by stacking $\nabla_w \log(\pi(s,a))$ over $\mathcal{S}\times\mathcal{A}$, i.e., $[\nabla_w \Pi_{\pi_w}](s,a) = \nabla_w \log(\pi_w(s,a))$, and $[\nabla_w \Pi_{\pi_w} \cdot Q_\pi] \in \mathbb{R}^{d_w|\mathcal{S}||\mathcal{A}|}$ is element-wise product between $\nabla_w \Pi_{\pi_w}$ and $Q_\pi$, i.e., $[\nabla_w \Pi_{\pi_w} \cdot Q_\pi](s,a) = \nabla_w \log(\pi_w(s,a))Q_\pi(s,a)$. Equation (15) implies that $\nabla_w Q_\pi$ is the mean of the accumulation of signal $C(s,a) = \gamma\mathbb{E}[Q_\pi(s',a')\nabla_w \log(\pi_w(s',a'))|s,a]$ with the discounted factor $\gamma$. Let $G_\pi = [Q_\pi^\top, \nabla_w Q^\top]^\top$. We have that $G_\pi$ satisfies the general Bellman equation in eq. (2) with $B$ and $M_\pi$ specified as

$$B = \left[\begin{array}{c} R \\ 0 \end{array}\right], \quad M_\pi = \left[\begin{array}{cc} \gamma\mathsf{P}_\pi & 0 \\ \gamma[\mathsf{P}_\pi \otimes \mathsf{I}_{d_w}]\mathrm{diag}(\nabla_w\Pi_{\pi_w}) & \gamma\mathsf{P}_\pi \otimes \mathsf{I}_{d_w} \end{array}\right], \tag{16}$$

where $\mathrm{diag}(\nabla_w\Pi_{\pi_w}) \in \mathbb{R}^{d_w|\mathcal{S}||\mathcal{A}|\times|\mathcal{S}||\mathcal{A}|}$ is obtained by arranging $\nabla_w \log(\pi(s,a)) \in \mathbb{R}^{d_w}$ diagonally. Without loss of generality, we assume that the score function is bounded [44, 47, 59], i.e., $\|\nabla_w \log(\pi_w(s,a))\|_2 \le C_\Pi$ for all $(s,a) \in \mathcal{S}\times\mathcal{A}$.

**Case III: Stochastic Value Gradient.** The stochastic value gradient (SVG) method combines advantages of model-based and model-free methods, in which both the estimated model and value function are updated to evaluate the policy gradient [12]. In the framework of SVG, the reward $r(s,a)$ is differentiable with respect to both $s \in \mathbb{R}^{d_s}$ an $a \in \mathbb{R}^{d_a}$, the stochastic policy takes the form $a = \pi(s,w) + \eta$, and the transition probability is modelled as $s' = f(s,a) + \xi$, where $\pi: \mathcal{S}\to\mathcal{A}$ and $f: \mathcal{S}\times\mathcal{A}\to\mathcal{S}$ are deterministic mappings, $w \in \mathbb{R}^{d_w}$ is the policy parameter, and $\eta \sim P(\eta)$ and $\xi \sim P(\xi)$ are noise variables. We abbreviate the partial differentiation using subscripts as $g_x \triangleq \partial g/\partial x$. The gradient of the Q-function w.r.t the policy parameter $w$ is given by [12]

$$\begin{aligned} Q_s &= (R_s + \Pi_s R_a) + \gamma(\Pi_s F_a + F_s)([\mathsf{P}_\pi \otimes \mathsf{I}_{d_s}]Q_s), \\ Q_a &= R_a + \gamma F_a[\mathsf{P}_\pi \otimes \mathsf{I}_{d_s}]Q_s + \gamma[\mathsf{P}_\pi \otimes \mathsf{I}_{d_a}]Q_a, \\ \nabla_w Q_\pi &= \Pi_w Q_a + \gamma[\mathsf{P}_\pi \otimes \mathsf{I}_{d_w}]\nabla_w Q_\pi, \end{aligned} \tag{17}$$

where $R_s \in \mathbb{R}^{d_s|\mathcal{S}||\mathcal{A}|}$ and $R_a \in \mathbb{R}^{d_a|\mathcal{S}||\mathcal{A}|}$ are vectors obtained via stacking partial derivatives $r_s(s,a) \in \mathbb{R}^{d_s}$ and $r_a(s,a) \in \mathbb{R}^{d_a}$ over $(s,a) \in \mathcal{S}\times\mathcal{A}$, and $\Pi_s = \mathrm{diag}(\pi_s) \in \mathbb{R}^{d_s|\mathcal{S}||\mathcal{A}|\times d_a|\mathcal{S}||\mathcal{A}|}$, $\Pi_w = \mathrm{diag}(\pi_w) \in \mathbb{R}^{d_w|\mathcal{S}||\mathcal{A}|\times d_a|\mathcal{S}||\mathcal{A}|}$, $F_s = \mathrm{diag}(f_s) \in \mathbb{R}^{d_s|\mathcal{S}||\mathcal{A}|\times d_s|\mathcal{S}||\mathcal{A}|}$, and $F_a = \mathrm{diag}(f_a) \in \mathbb{R}^{d_a|\mathcal{S}||\mathcal{A}|\times d_s|\mathcal{S}||\mathcal{A}|}$ are Jacobian matrices. Consider GVF defined as $G_\pi = [Q_s^\top, Q_a^\top, \nabla_w Q_\pi^\top]^\top$. Consider the normalized setting in which $\Pi_s F_a + F_s = \mathbf{I}$. Then $G_\pi$ satisfies the general Bellman equation in eq. (2) with $B$ and $M_\pi$ specified by

$$B = \left[\begin{array}{c} R_s + \Pi_s R_a \\ R_a \\ \Pi_w Q_a \end{array}\right], \quad M_\pi = \left[\begin{array}{ccc} \gamma[\mathsf{P}_\pi \otimes \mathsf{I}_s] & 0 & 0 \\ \gamma F_a[\mathsf{P}_\pi \otimes \mathsf{I}_s] & \gamma[\mathsf{P}_\pi \otimes \mathsf{I}_a] & 0 \\ 0 & \Pi_w & \gamma[\mathsf{P}_\pi \otimes \mathsf{I}_{d_w}] \end{array}\right]. \tag{18}$$

We consider the setting in which $\|f_a(s,a)\|_F \le C_a$ and $\|\pi_w(s,a)\|_F \le C_w$ for all $(s,a) \in \mathcal{S}\times\mathcal{A}$. We make the following "non-expansive" assumption for the transition matrix $\gamma_1(s,a) = \gamma(\Pi_s F_a + F_s)$.

**Assumption 3** (Normalization). *For any $v \in \mathbb{R}^{d_s \times 1}$, we have $\|(\Pi_s F_a + F_s)v\|_{U_\pi} \leq \|v\|_{U_\pi}$.*

Assumption 3 is the minimum requirement to guarantee the value of $Q_s$ to be bounded. It can be satisfied by selecting appropriate policy class $\pi_w$ and model approximation $f$ together with the feature design of both state $s$ and action $a$.

**Case IV: Option Learning.** In the option framework [49], an option is defined as $(\pi_o, \lambda_o, \mathcal{O})$, where $\pi_o : \mathcal{S} \times \mathcal{A} \to [0,1]$ is an intra-option policy, $\lambda_o : \mathcal{S} \to [0,1]$ is a termination function, and $\mathcal{O}$ is the option set. In this framework, the policy $\pi$ is defined over the option-state space, i.e., $\pi : \mathcal{O} \times \mathcal{S} \to [0,1]$. At time step $t$, an agent at state $s_t$ either terminates the previous option $o_{t-1}$ with probability $\lambda_{o_{t-1}}(s_t)$ and initiates a new option $o_t$ according to policy $\pi(\cdot|s_t)$, or proceeds with the previous option $o_{t-1}$ with probability $1 - \lambda_{o_{t-1}}(s_t)$ and sets $o_t = o_{t-1}$. Then an action $a_t$ is selected according to $\pi_{o_t}(\cdot|s_t)$. The agent receive a reward $r(s_t, a_t)$. Similar to regular MDP, here we define state-option value function as $Q_\pi(s, o) = \int_a \pi(a|s)Q_\pi(s, o, a)da$, where $Q_\pi(s, o, a) = \mathbb{E}[\sum_{t=0}^\infty \gamma^t r(s_t, a_t)|s_0 = s, a_0 = a, o_0 = o]$. We consider evaluate the state-option value function $Q_\pi(s, o)$. [71] shows that

$$Q_\pi^{\mathcal{O}} = R + \gamma P_\pi^{\lambda,\mathcal{O}} Q_\pi^{\mathcal{O}}, \tag{19}$$

where $[P_\pi^{\lambda,\mathcal{O}}]((s, o), (s', o')) = [(1 - \lambda_o(s'))\mathbb{1}_{o'=o} + \lambda_o(s')\pi(o'|s')]\mathsf{P}(s'|s, o)$, where $\mathsf{P}(s'|s, o) = \int_a \pi_o(a|s)\mathsf{P}(s'|s, a)$. Let $Y_\pi(x) = Q_\pi(s, o)$, $C(x_t) = r(s_t, a_t)$ and $m(x, x_{t+1}) = \gamma$, it can be checked that $Y_\pi$ satisfies the general Bellman equation defined in Section 2.1.

### C.2 Examples of Backward GVFs

The backward GVF in Definition 2 also arises in the following important RL applications.

**Case IV: Anomaly Detection.** [70] has systemically discussed the application of retrospective knowledge in anomaly detection. Let $i(s, a)$ be the cost that an agent consumes when taking action $a$ at state $s$, and $e_\pi(s, a)$ be the cost that an agent is expected to consume given the current status when following a predefined policy $\pi$. If the actual cost of the agent deviates too much from $e_\pi$, the agent may likely encounter anomalous events. For simplicity, we consider the setting when $\gamma(s, a) = \gamma$. It can be shown that $e_\pi$ satisfies the following equation

$$e_\pi = i + \gamma U_\pi^{-1} \mathsf{P}_\pi^\top U_\pi e_\pi. \tag{20}$$

Clearly, eq. (20) satisfies the general backward Bellman equation in eq. (4) by letting $B = i$, $M_\pi = \gamma \mathsf{P}_\pi$, and $\hat{G}_\pi = e_\pi$.

**Case V: Gradient of Logarithmic Stationary Distributions.** In the policy parameterization setting, the gradient of logarithmic stationary distribution $\nabla_w \log \mu_\pi(s, a)$ has been used in policy gradient estimation [18, 61, 26] and maximum entropy exploration [11]. It has been shown in [26, 61] that $\nabla_w \log \mu_\pi(s, a)$ satisfies the following equation

$$\Psi_\pi = \nabla_w \Pi_{\pi_w} + U_\pi^{-1}[\mathsf{P}_\pi^\top \otimes \mathsf{I}_{d_w}]U_\pi \Psi_\pi, \tag{21}$$

where $\Psi_\pi$ is obtained via stacking $\nabla_w \log(\mu_\pi(s, a))$ over $\mathcal{S} \times \mathcal{A}$, i.e., $[\Psi_\pi](s, a) = \nabla_w \log(\mu_{\pi_w}(s, a))$. Here, $\nabla_w \log \mu_\pi$ can be viewed as a backward accumulation of the signal $C(s, a) = \nabla_w \log(\pi(s, a))$ with the discounted factor $\gamma = 1$. Define the backward GVF as $\hat{G}_\pi = \Psi_\pi$. It is clear that $\hat{G}_\pi$ satisfies the general backward Bellman equation in eq. (4) with $B$ and $M_\pi$ specified by

$$B = \nabla_w \Pi_{\pi_w}, \qquad M_\pi = \mathsf{P}_\pi^\top \otimes \mathsf{I}_{d_w}. \tag{22}$$

Note that since $\gamma_{\max} = 1$ in the general Bellman equation in eq. (21), the result in Proposition 1 may not hold in such a setting, i.e., GBO may not be a contraction here. However, as we will show in appendix H, when the base matrix $\Phi$ satisfies the "non-constant parameterization" assumption, we can establish results similar to Proposition 2 and Theorem 3 for the evaluation of $\nabla_w \log \mu_\pi$.

## D   Gradient Temporal Difference Learning (GTD)

The GTD algorithm has been used for GVF evaluation in [48, 37]. So far, only the asymptomatic convergence (not the convergence rate) has been studied in [37]. In this section, we present the

---
**Algorithm 2** GTD
---
**Initialize:** Approximator parameters $w_0$, and $\theta_0$
   **for** $t = 0, \cdots, T - 1$ **do**
      Obtain sample $(s_t, a_t, B_t, s'_t) \sim \mathcal{D}$ and $a'_t \sim \pi(\cdot|s'_t)$
      $w_{t+1} = w_t - \beta_t(g(x_t, \theta_t) - l(x_t, w_t))$
          **forward GVF:** $l(x_t, w_t) = \phi(s_t, a_t)^\top \phi(s_t, a_t) w_t$
          **backward GVF:** $l(x_t, w_t) = \phi(s'_t, a'_t)^\top \phi(s'_t, a'_t) w_t$
      $\theta_{t+1} = \Gamma_{R_\theta} \left( \theta_t + \alpha_t(g(x_t, \theta_t) - h(x_t, \theta_t)) \right)$
          **forward GVF:** $h(x_t, w_t) = \phi(s'_t, a'_t)^\top m(x_t) \phi(s_t, a_t) w_t$
          **backward GVF:** $h(x_t, w_t) = \phi(s_t, a_t)^\top \hat{m}(x_t) \phi(s'_t, a'_t) w_t$
   **end for**
---

GTD algorithm for GVF evaluation and characterize the finite-time convergence rate for GTD. We define the dimension of parameter $\theta$ as $d_g = \sum_{i=1}^{k} K_i d_i$. In the sequel, we denote the MSPBE with parameter $\theta$ as $J(\theta)$. Note that the MSPBE in eq. (5) can be rewritten as

$$J(\theta) = \frac{1}{2} \left\| \mathbb{E}_D[g(x, \theta)] \right\|_{C^{-1}}^2, \ \ C \triangleq \begin{cases} \mathbb{E}_D[(\phi(s, a)^\top \phi(s, a))] & \text{(forward GVF)} \\ \mathbb{E}_D[(\phi(s', a')^\top \phi(s', a'))] & \text{(backward GVF)} \end{cases}, \tag{23}$$

where

$$g(x, \theta) \triangleq \begin{cases} -\phi(s, a)^\top (B(s, a) + m(x)\phi(s', a')\theta - \phi(s, a)\theta) & \text{(forward GVF)} \\ -\phi(s', a')^\top (B(s', a') + \hat{m}(x)\phi(s, a)\theta - \phi(s', a')\theta) & \text{(backward GVF)} \end{cases},$$

where the matrices $m(\cdot)$ and $\hat{m}(\cdot)$ are defined in eq. (8). The gradient of $J(\theta)$ is given as

$$-\nabla J(\theta) = \mathbb{E}_D[g(x, \theta) - h(x, \theta)], \ \ h(x, \theta) \triangleq \begin{cases} \phi(s', a')^\top m(x)\phi(s, a)w(\theta) & \text{(forward GVF)} \\ \phi(s, a)^\top \hat{m}(x)\phi(s', a')w(\theta) & \text{(backward GVF)} \end{cases}$$

in which $w(\theta) = C^{-1}\mathbb{E}_D[g(x, \theta)] \in \mathbb{R}^{d_g}$. Note that we can not estimate $\nabla J(\theta)$ directly due to the "double sampling" issue, i.e., $w(\theta)$ cannot be estimated via sampling. In GTD, an auxiliary parameter $w_t$ is introduced, which is updated simultaneously with $\theta_t$ to approximate $w(\theta_t)$ [45, 23]. We present the update of GTD in Algorithm 2.

### D.1 Convergence Rate of GTD

We make the following assumptions, which have also been adopted in the convergence analysis of GTD in the canonical value function evaluation setting [62, 60, 45, 23].

**Assumption 4.** *In both forward and backward GVF evaluation settings, the matrix $C$ in eq. (23) is non-singular.*

**Assumption 5.** *We define the matrix $\hat{A}$ in the following way: (1) in the forward GVF evaluation setting: $\hat{A} = \mathbb{E}_{\mathcal{D} \cdot \pi}[\phi(s, a)^\top (m(x)\phi(s', a') - \phi(s, a))]$; (2) in the backward GVF evaluation setting, $\hat{A} = \mathbb{E}_{\mathcal{D} \cdot \pi}[\phi(s', a')(\hat{m}(x)\phi(s, a) - \phi(s', a'))]$. We require $\hat{A}$ to be non-singular in both the forward and backward GVF settings.*

We define the optimal point $\bar{\theta}^*$ for GTD as

$$\langle \nabla J(\bar{\theta}^*), \theta - \bar{\theta}^* \rangle \geq 0, \quad \forall \theta \in R_\theta,$$

which is the optimality condition for minimizing $J(\theta)$. The following theorem characterizes the convergence rate of GTD to $\bar{\theta}^*$.

**Theorem 3.** *Consider the GTD update in Algorithm 2. In both the forward and backward GVF evaluation settings, suppose Assumption 4-5 hold. Let the stepsize $\alpha_t = \Theta(t^{-1})$ and $\beta_t = \Theta(t^{-1})$. We have*

$$\mathbb{E}\left[ \left\| \theta_T - \bar{\theta}^* \right\|_2^2 \right] \leq \mathcal{O}\left( \frac{\left\| \theta_0 - \bar{\theta}^* \right\|_2^2}{T^2} \right) + \mathcal{O}\left( \frac{1}{\lambda_G'^2 T} \right),$$

*where $\lambda_G' > 0$ is the conditional number of GTD defined in eq. (71) of Appendix I.*

Theorem 1 shows that GTD converges to the globally optimal point $\bar{\theta}^*$ at a rate of $\mathcal{O}(1/T)$. The convergence speed of $\theta_t$ depends on the conditional number $\lambda_G'$, which decreases as $\lambda_G'$ decreases. Differently from the conditional number $\lambda_G$ of GenTD, which has a guaranteed lower bound from zero as given in Proposition 2, there exists no guaranteed lower bound for $\lambda_G'$ even in the canonical value function evaluation setting. Thus, the converge speed of GTD could be very slow as $\lambda_G'$ could be arbitrarily small.

### D.2 Global Optimum of GTD and Proof of Example 1

For simplicity, we consider scenarios when the function approximation class $\mathcal{F}_\Phi$ is complete. We show that the global optimum of GTD exhibits very different properties in the forward and backward GVF evaluation settings.

We first show that in the forward GVF evaluation setting, the global optimum $\Phi\bar{\theta}^*$ of GTD equals the ground truth GVF. Since the function space $\mathcal{F}_\Phi$ is complete, there exists a parameter $\theta_{\text{true}} \in R_\theta$ such that $\Phi\theta_{\text{true}} = G_\pi$, which implies $J(\theta_{\text{true}}) = 0$. Since $J(\theta) \geq 0$ for all $\theta \in R_\theta$ and $J(\theta)$ is strongly-convex, $J(\theta) = 0$ if and only if $\theta = \bar{\theta}^*$, which implies $\theta_{\text{true}} = \bar{\theta}^*$.

In the backward GVFs evaluation setting, we provide an example (see Example 1 in Section 4) to show that GTD can fail to learn the ground truth $G_\pi$ even if the function class $\mathcal{F}_\Phi$ is complete. We next present the proof for such an example.

**Proof of Example 1.** The backward value function can be obtained as follows

$$\bar{V} = U^{-1}(I - \gamma \mathsf{P}^\top)^{-1} \mathsf{P}^\top U R = [8.1555, 9.0389, 9.0184]^\top.$$

The fixed point of GTD is given by

$$\bar{\theta}^* = \bar{A}^{-1}\bar{b},$$

where

$$\bar{A} = \gamma \Phi^\top \mathsf{P}^\top \bar{D}'\Phi - \Phi^\top \bar{D}'\Phi, \qquad \bar{b} = \Phi^\top \mathsf{P}^\top \bar{D}R, \tag{24}$$

where $\bar{D} = \text{diag}(D)$, $\bar{D}' = \text{diag}(D')$ and $D'^\top = D^\top P$. Also note that the base matrix $\Phi = [8.1555, 9.0389, 9.0184]^\top$ and the off-policy sampling distribution $D = [1/3, 1/3, 1/3]^\top$. We can obtain

$$\bar{A} = -9.9422, \qquad \bar{b} = 5.9904,$$

which implies

$$\bar{\theta}^* = \frac{-b}{A} = 0.6025. \tag{25}$$

Note that the perfect base matrix $[8.1555, 9.0389, 9.0184]^\top$ can fully represent $\bar{V}_\pi$, with parameter $\theta_{\text{true}} = 1$. However, eq. (25) shows that the global optimum of GTD $\bar{\theta}^* \neq \theta_{\text{true}}$, which introduces a non-zero approximation error:

$$\left\|\Phi\bar{\theta}^* - \bar{V}\right\|_D = 3.7848.$$

$\square$

The above example demonstrates a drawback of GTD, which can fail to learn the ground truth $G_\pi$ even if the function class $\mathcal{F}_\Phi$ is complete. Such an issue does not occur for the GenTD algorithm proposed in this paper, which converges to the ground truth as guaranteed by Theorem 2 in Section 4.

## E   Proofs of Propositions 1 and 2

### E.1   Supporting Lemmas

We provide the following lemmas, which are useful for the proofs of Propositions 1 and 2.

**Lemma 1.** *For any $v \in \mathbb{R}^{d|\mathcal{S}||\mathcal{A}|}$, we have $\|[\mathsf{P}_\pi \otimes \mathsf{I}_d]v\|_{U_\pi} \leq \|v\|_{U_\pi}$.*

*Proof.* Consider the square of $\left\|[\mathsf{P}_\pi \otimes \mathsf{I}_d]v\right\|_{U_\pi}$. We have

$$\|[\mathsf{P}_\pi \otimes \mathsf{I}_d]v\|_{U_\pi}^2 = v^\top [\mathsf{P}_\pi^\top \otimes \mathsf{I}_d]U_\pi[\mathsf{P}_\pi \otimes \mathsf{I}_d]v$$

$$= \sum_{i=1}^{|\mathcal{S}||\mathcal{A}|} \mu_\pi(i) \left\| \sum_{j=1}^{|\mathcal{S}||\mathcal{A}|} \mathsf{P}_\pi(j|i)v_j \right\|_2^2$$

$$\leq \sum_{i=1}^{|\mathcal{S}||\mathcal{A}|} \mu_\pi(i) \sum_{j=1}^{|\mathcal{S}||\mathcal{A}|} \mathsf{P}_\pi(j|i) \|v_j\|_2^2$$

$$= \sum_{i=1}^{|\mathcal{S}||\mathcal{A}|} \sum_{j=1}^{|\mathcal{S}||\mathcal{A}|} \mu_\pi(i)\mathsf{P}_\pi(j|i) \|v_j\|_2^2$$

$$= \sum_{j=1}^{|\mathcal{S}||\mathcal{A}|} \mu_\pi(i) \|v_j\|_2^2$$

$$= \|v\|_{U_\pi}^2,$$

where the first inequality follows from Jensen's inequality and the fourth equality follows from the property of the stationary distribution $\mu_\pi^\top \mathsf{P}_\pi = \mu_\pi^\top$. $\qquad\square$

We provide the follow lemma to characterize a similar property in backward GVF evaluation setting.

**Lemma 2.** *For any $v \in \mathbb{R}^{d|\mathcal{S}||\mathcal{A}|}$, we have* $\left\|U_\pi^{-1}[\mathsf{P}_\pi^\top \otimes \mathsf{I}_d]U_\pi v\right\|_{U_\pi} \leq \|v\|_{U_\pi}$.

*Proof.* Consider the square of $\left\|U_\pi^{-1}[\mathsf{P}_\pi^\top \otimes \mathsf{I}_d]U_\pi v\right\|_{U_\pi}$. We have

$$\left\|U_\pi^{-1}[\mathsf{P}_\pi \otimes \mathsf{I}_d]U_\pi v\right\|_{U_\pi}^2 = v^\top [U_\pi^{-1}[\mathsf{P}_\pi^\top \otimes \mathsf{I}_d]U_\pi]^\top U_\pi [U_\pi^{-1}[\mathsf{P}_\pi^\top \otimes \mathsf{I}_d]U_\pi]v$$

$$= \sum_{j=1}^{|\mathcal{S}||\mathcal{A}|} \mu_\pi(j) \left\| \sum_{i=1}^{|\mathcal{S}||\mathcal{A}|} \frac{\mu_\pi(i)\mathsf{P}_\pi(j|i)}{\mu_\pi(j)}v(i) \right\|_2^2$$

$$\leq \sum_{j=1}^{|\mathcal{S}||\mathcal{A}|} \mu_\pi(j) \sum_{i=1}^{|\mathcal{S}||\mathcal{A}|} \frac{\mu_\pi(i)\mathsf{P}_\pi(j|i)}{\mu_\pi(j)} \|v(i)\|_2^2$$

$$= \sum_{j=1}^{|\mathcal{S}||\mathcal{A}|} \mu_\pi(i) \|v(i)\|_2^2 \sum_{i=1}^{|\mathcal{S}||\mathcal{A}|} \mathsf{P}_\pi(j|i)$$

$$= \sum_{j=1}^{|\mathcal{S}||\mathcal{A}|} \mu_\pi(i) \|v(i)\|_2^2$$

$$= \|v\|_{U_\pi}^2,$$

where the first inequality follows from the Jensen's inequality. $\qquad\square$

### E.2 Proof of Proposition 1

We first consider the **forward GBO setting**. Recall the following definition of GBO $\mathcal{T}_{G,\pi}$ in eq. (2)

$$G_\pi = \mathcal{T}_{G,\pi}G_\pi = B + M_\pi G_\pi,$$

where

$$B = \begin{bmatrix} B_1 \\ B_2 \\ \vdots \\ B_k \end{bmatrix}, \quad M_\pi = \begin{bmatrix} \gamma_1[\mathsf{P}_\pi \otimes \mathsf{I}_{d_1}] & 0 & \cdots & 0 \\ A_{2,1} & \gamma_2[\mathsf{P}_\pi \otimes \mathsf{I}_{d_2}] & \cdots & 0 \\ \vdots & \vdots & & \vdots \\ A_{k,1} & A_{k,2} & \cdots & \gamma_k[\mathsf{P}_\pi \otimes \mathsf{I}_{d_k}] \end{bmatrix}.$$

Let $G'_\pi, G''_\pi \in \mathbb{R}^{|\mathcal{S}||\mathcal{A}|\sum_{i=1}^k K_i d_i}$ be two vectors, and let $\Delta_G = [\Delta_1, \cdots, \Delta_k]$, where $\Delta_i = G'_{\pi,i} - G''_{\pi,i}$. We have

$$\mathcal{T}_\pi G'_\pi - \mathcal{T}_\pi G''_\pi = M_\pi \Delta_G = \begin{bmatrix} \gamma_1[\mathsf{P}_\pi \otimes \mathsf{I}_{d_1}]\Delta_1 \\ A_{2,1}\Delta_1 + \gamma_2[\mathsf{P}_\pi \otimes \mathsf{I}_{d_2}]\Delta_2 \\ \vdots \\ \sum_{j=1}^{k-1} A_{k,j}\Delta_j + \gamma_k[\mathsf{P}_\pi \otimes \mathsf{I}_{d_k}]\Delta_k \end{bmatrix}. \tag{26}$$

Recall that $A_{i,j}$ is bounded for all $i, j$. Thus, there exists a constant $0 < C_A < \infty$ such that $\|A_{i,j}\|_{U_\pi} \le C_A$ for all $i, j$. Without loss of generality, we assume $C_A > 1$. Let $\alpha$ be the solution of the following matrix function

$$Fx = f, \tag{27}$$

where $F \in \mathbb{R}^{k \times k}$ and $f \in \mathbb{R}^k$ are specified as

$$F = \begin{bmatrix} -\frac{1-\gamma}{2} & C_A & C_A & \cdots & C_A & C_A \\ 0 & -\frac{1-\gamma}{2} & C_A & \cdots & C_A & C_A \\ \vdots & \vdots & & \vdots & & \\ 0 & 0 & 0 & \cdots & -\frac{1-\gamma}{2} & C_A \\ 1 & 1 & 1 & \cdots & 1 & 1 \end{bmatrix}, \qquad f = \begin{bmatrix} 0 \\ 0 \\ \vdots \\ 0 \\ 1 \end{bmatrix}.$$

It can be checked that the solution of eq. (27) is strictly positive, i.e., if $F\alpha = f$, then we have $\alpha_l > 0$ for $1 \le l \le k$. Recalling the definition of $\|\cdot\|_{\mu_\pi,\alpha}$– norm, we have

$$\|M_\pi \Delta_G\|_{\mu_\pi,\alpha}$$
$$= \gamma_1\alpha_1 \|[\mathsf{P}_\pi \otimes \mathsf{I}_{d_1}]\Delta_1\|_{\mu_\pi} + \alpha_2 \|A_{2,1}\Delta_1 + \gamma_2[\mathsf{P}_\pi \otimes \mathsf{I}_{d_2}]\Delta_2\|_{\mu_\pi}$$
$$+ \cdots + \alpha_k \left\| \sum_{j=1}^{k-1} A_{k,j}\Delta_j + \gamma_k[\mathsf{P}_\pi \otimes \mathsf{I}_{d_k}]\Delta_k \right\|_{\mu_\pi}$$
$$\le \gamma_1\alpha_1 \|[\mathsf{P}_\pi \otimes \mathsf{I}_{d_1}]\Delta_1\|_{\mu_\pi} + \alpha_2 \|A_{2,1}\Delta_1\|_{\mu_\pi} + \cdots + \alpha_k \|A_{k,1}\Delta_1\|_{\mu_\pi}$$
$$+ \gamma_2\alpha_2 \|[\mathsf{P}_\pi \otimes \mathsf{I}_{d_2}]\Delta_2\|_{\mu_\pi} + \alpha_3 \|A_{3,2}\Delta_2\|_{\mu_\pi} + \cdots + \alpha_k \|A_{k,2}\Delta_2\|_{\mu_\pi}$$
$$+ \cdots$$
$$+ \gamma_k\alpha_k \|[\mathsf{P}_\pi \otimes \mathsf{I}_{d_k}]\Delta_k\|_{\mu_\pi}$$
$$\le \left( \gamma\alpha_1 + C_A \sum_{i=2}^k \alpha_i \right) \|\Delta_1\|_{\mu_\pi} + \left( \gamma\alpha_2 + C_A \sum_{i=3}^k \alpha_i \right) \|\Delta_2\|_{\mu_\pi} + \cdots + \gamma\alpha_k \|\Delta_k\|_{\mu_\pi}$$
$$\le \frac{1+\gamma}{2}\alpha_1 \|\Delta_1\|_{\mu_\pi} + \frac{1+\gamma}{2}\alpha_2 \|\Delta_2\|_{\mu_\pi} + \cdots + \gamma\alpha_k \|\Delta_k\|_{\mu_\pi}$$
$$\le \frac{1+\gamma}{2} \left( \sum_{i=1}^k \alpha_i \|\Delta_i\|_{\mu_\pi} \right)$$
$$= \frac{1+\gamma}{2} \|\Delta_G\|_{\mu_\pi,\alpha}, \tag{28}$$

where the first inequality follows from the triangle inequality, the second inequality follows from the fact that $A_{i,j}$ is bounded and Lemma 1, and the third inequality follows from the definition of $\gamma$ and the fact that $\alpha$ is the solution of eq. (27). Obviously, eq. (28) implies the following property,

$$\|\mathcal{T}_\pi G'_\pi - \mathcal{T}_\pi G''_\pi\|_{\mu_\pi,\alpha} \le \frac{1+\gamma}{2} \|G'_\pi - G''_\pi\|_{\mu_\pi,\alpha},$$

which completes the proof in the forward GBO evaluation setting.

We next consider the **backward GBO setting**, where $\hat{\mathcal{T}}_{G,\pi}$ is defined in eq. (3). Following steps similar to those from eq. (26) – eq. (28), we can obtain

$$\left\| \hat{\mathcal{T}}_\pi G_\pi - \hat{\mathcal{T}}_\pi G'_\pi \right\|_{\mu_\pi,\alpha} = \|M_\pi \Delta_G\|_{\mu_\pi,\alpha}$$

$$= \gamma_1 \alpha_1 \left\| U_{\pi,1}^{-1}[\mathsf{P}_\pi \otimes \mathsf{I}_{d_1}]U_{\pi,1}\Delta_1 \right\|_{\mu_\pi} + \alpha_2 \left\| A_{2,1}\Delta_1 + \gamma_2 U_{\pi,2}^{-1}[\mathsf{P}_\pi \otimes \mathsf{I}_{d_2}]U_{\pi,2}\Delta_2 \right\|_{\mu_\pi} + \cdots$$

$$+ \alpha_k \left\| \sum_{j=1}^{k-1} A_{k,j}\Delta_j + \gamma_k U_{\pi,k}^{-1}[\mathsf{P}_\pi \otimes \mathsf{I}_{d_k}]U_{\pi,k}\Delta_k \right\|_{\mu_\pi}$$

$$\leq \gamma_1 \alpha_1 \left\| U_{\pi,1}^{-1}[\mathsf{P}_\pi \otimes \mathsf{I}_{d_1}]U_{\pi,1}\Delta_1 \right\|_{\mu_\pi} + \alpha_2 \left\| A_{2,1}\Delta_1 \right\|_{\mu_\pi} + \cdots + \alpha_k \left\| A_{k,1}\Delta_1 \right\|_{\mu_\pi}$$

$$+ \gamma_2 \alpha_2 \left\| U_{\pi,2}^{-1}[\mathsf{P}_\pi \otimes \mathsf{I}_{d_2}]U_{\pi,2}\Delta_2 \right\|_{\mu_\pi} + \alpha_3 \left\| A_{3,2}\Delta_2 \right\|_{\mu_\pi} + \cdots + \alpha_k \left\| A_{k,2}\Delta_2 \right\|_{\mu_\pi}$$

$$+ \cdots$$

$$+ \gamma_k \alpha_k \left\| U_{\pi,k}^{-1}[\mathsf{P}_\pi \otimes \mathsf{I}_{d_k}]U_{\pi,k}\Delta_k \right\|_{\mu_\pi}$$

$$\leq \left( \gamma \alpha_1 + C \sum_{i=2}^{k} \alpha_i \right) \left\| \Delta_1 \right\|_{\mu_\pi} + \left( \gamma \alpha_2 + C \sum_{i=3}^{k} \alpha_i \right) \left\| \Delta_2 \right\|_{\mu_\pi} + \cdots + \gamma \alpha_k \left\| \Delta_k \right\|_{\mu_\pi}$$

$$\leq \frac{1+\gamma}{2} \alpha_1 \left\| \Delta_1 \right\|_{\mu_\pi} + \frac{1+\gamma}{2} \alpha_2 \left\| \Delta_2 \right\|_{\mu_\pi} + \cdots + \gamma \alpha_k \left\| \Delta_k \right\|_{\mu_\pi}$$

$$\leq \frac{1+\gamma}{2} \left( \sum_{i=1}^{k} \alpha_i \left\| \Delta_i \right\|_{\mu_\pi} \right)$$

$$= \frac{1+\gamma}{2} \left\| \Delta_G \right\|_{\mu_\pi,\alpha}, \tag{29}$$

where the first inequality follows from the triangle inequality, the second inequality follows from the fact that $A_{i,j}$ is bounded and Lemma 2, and the third inequality follows from the definition of $\gamma$ and the fact that $\alpha$ is the solution of eq. (27). Equation (29) implies the following

$$\left\| \hat{\mathcal{T}}_\pi G'_\pi - \hat{\mathcal{T}}_\pi G''_\pi \right\|_{\mu_\pi,\alpha} \leq \frac{1+\gamma}{2} \left\| G'_\pi - G''_\pi \right\|_{\mu_\pi,\alpha},$$

which completes the proof in the backward GBO evaluation setting.

### E.3 Proof of Proposition 2

We first consider the **forward GFV setting**. Recall the linear function approximation of $G_\pi$ is given by

$$G(\theta) = \Phi\theta,$$

where

$$\Phi = \begin{bmatrix} \Phi_1 \otimes \mathsf{I}_{d_1} & 0 & \cdots & 0 \\ 0 & \Phi_2 \otimes \mathsf{I}_{d_2} & \cdots & 0 \\ \vdots & \vdots & & \vdots \\ 0 & 0 & \cdots & \Phi_k \otimes \mathsf{I}_{d_k} \end{bmatrix}, \quad \theta = \begin{bmatrix} \mathrm{vec}(\theta_1^\top) \\ \mathrm{vec}(\theta_2^\top) \\ \vdots \\ \mathrm{vec}(\theta_k^\top) \end{bmatrix}.$$

Folloing the definition of $g(\theta)$ in eq. (7), we have

$$-g(\theta) = \Phi^\top U_\pi(\mathcal{T}_{G,\pi}G(\theta) - G(\theta))$$
$$= \Phi^\top U_\pi((M_\pi - I)\Phi\theta + B)$$
$$= G\theta + g,$$

where $G = \Phi^\top U_\pi(M_\pi - I)\Phi$ and $g = \Phi^\top U_\pi B$. Since the monotonicity depends only on the matrix $G$, we next proceed to show that $G$ is Hurwitz. For the matrix $G$, we have

$$G = \Phi^\top U_\pi(M_\pi - I)\Phi = \begin{bmatrix} A_1 & 0 & \cdots & 0 \\ N_{2,1} & A_2 & \cdots & 0 \\ \vdots & \vdots & & \vdots \\ N_{k,1} & N_{k,2} & \cdots & A_k \end{bmatrix}, \tag{30}$$

where $A_i = [\Phi_i^\top U_{\pi,i}(\gamma_i \mathsf{P}_\pi - I)\Phi_i] \otimes \mathsf{I}_{d_i}$ and $N_{i,j}$ is a matrix that depends on $\Phi_i$, $\Phi_j$, $\mathsf{P}_\pi$ and $\mu_\pi$. We have the following equations hold:

$$\mathrm{eig}(G) = \{\mathrm{eig}(A_1), \cdots, \mathrm{eig}(A_k)\}, \tag{31}$$

$$\mathrm{eig}(A_i) = \mathrm{eig}(\Phi_i^\top U_{\pi,i}(\gamma_i \mathsf{P}_\pi - I)\Phi_i), \tag{32}$$

$$\max\{\mathrm{eig}(\Phi_i^\top U_{\pi,i}(\gamma_i \mathsf{P}_\pi - I)\Phi_i)\} = -(1 - \gamma)\zeta_i, \tag{33}$$

where $\zeta_i$ is defined in Proposition 2, the first equation follows because the eigenvalue of a matrix is determined by the eigenvalues of its diagonal block matrices [13], the second equation follows from the fact that $\mathrm{eig}(M \otimes \mathsf{I}_d) = \mathrm{eig}(M)$ for any matrix $M$ and positive integer $d$, and the last follows from Lemma 1 and Lemma 3 in [2]. Combining eq. (31)–(33), we can obtain equation

$$\max\{\mathrm{eig}(G)\} \le -(1 - \gamma)\min_i \zeta_i = -\lambda_G < 0, \tag{34}$$

which completes the proof in the forward GVF setting.

We next consider the **backward GVF setting**. Following the steps similar to those for deriving eq. (30), we can obtain $-g(\theta) = \hat{G}\theta + \hat{g}$, where $\hat{g} = \Phi^\top P_\pi^\top U_\pi B$. For the matrix $\hat{G}$, we have

$$\hat{G} = \Phi^\top P_\pi^\top (\hat{M}_\pi - I)U_\pi \Phi = \begin{bmatrix} \hat{A}_1 & 0 & \cdots & 0 \\ \hat{N}_{2,1} & \hat{A}_2 & \cdots & 0 \\ \vdots & \vdots & & \vdots \\ \hat{N}_{k,1} & \hat{N}_{k,2} & \cdots & \hat{A}_k \end{bmatrix}, \tag{35}$$

where $\hat{A}_i = [\Phi_i^\top(\gamma_i \mathsf{P}_\pi^\top - I)U_{\pi,i}\Phi_i] \otimes \mathsf{I}_{d_i}$ and $\hat{N}_{i,j}$ is a matrix that depends on $\Phi_i$, $\Phi_j$, $\mathsf{P}_\pi$ and $\mu_\pi$. Following the steps similar to those in eq. (31)–(34) and using the result in the verification of item (c) in Assumption 2 in [70], we have

$$\max\{\mathrm{eig}(G)\} \le -(1 - \gamma)\min_i \zeta_i = -\lambda_G < 0,$$

which completes the proof in the backward GVF setting.

# F   Proof of Theorem 1

## F.1   Supporting Lemmas

We first develop the property for the update of $w_\rho$ in Algorithm 1. Given a sample $(s_t, a_t, B_t, s_t') \sim \mathcal{D}$ and $a_t' \sim \pi(\cdot|s_t')$, we introduce the following definitions.

$$P_t = \begin{bmatrix} \psi_t^\top \psi_t & (\psi_t - \psi_t')\psi_t^\top & 0 \\ -\psi_t(\psi_t^\top - \psi_t'^\top) & 0 & \psi_t \\ 0 & -\psi_t^\top & 1 \end{bmatrix}, \qquad p_t = \begin{bmatrix} 0 \\ 0 \\ 1 \end{bmatrix}.$$

Consider the matrix $P_t$ and vector $p_t$, we have the following holds

$$\|P_t\|_F^2 = \|\psi_t^\top \psi_t\|_F^2 + 2\|(\psi_t - \psi_t')\psi_t^\top\|_F^2 + 2\|\psi_t\|_F^2 + 1$$
$$\le 9C_\psi^4 + 2C_\psi^2 + 1, \tag{36}$$

where $C_\psi$ is the upper bound on the feature fector $\psi(\cdot)$, i.e., $\|\psi(s, a)\|_2 \le C_\psi$ for all $(s, a) \in \mathcal{S} \times \mathcal{A}$, which implies $\|P_t\|_F \le C_P$, where

$$C_P = \sqrt{9C_\psi^4 + 2C_\psi^2 + 1}. \tag{37}$$

For the vector $p_t$, it can be checked easily that $\|p_t\|_2 \le 1$.

We also define $P = \mathbb{E}_{\mathcal{D}\cdot\pi}[P_t]$ and $p = \mathbb{E}_{\mathcal{D}\cdot\pi}[p_t]$, i.e.,

$$P = \begin{bmatrix} \mathbb{E}_{\mathcal{D}\cdot\pi}[\psi^\top \psi] & \mathbb{E}_{\mathcal{D}\cdot\pi}[(\psi - \psi')\psi^\top] & 0 \\ -\mathbb{E}_{\mathcal{D}\cdot\pi}[\psi(\psi^\top - \psi'^\top)] & 0 & \mathbb{E}_{\mathcal{D}\cdot\pi}[\psi] \\ 0 & -\mathbb{E}_{\mathcal{D}\cdot\pi}[\psi^\top] & 1 \end{bmatrix}, \qquad p = \begin{bmatrix} 0 \\ 0 \\ 1 \end{bmatrix}.$$

Note that

$$
\begin{bmatrix}
\nabla_{w_f} L(\hat{\rho}, \hat{f}, \eta) \\
\nabla_{w_\rho} L(\hat{\rho}, \hat{f}, \eta) \\
\nabla_\eta L(\hat{\rho}, \hat{f}, \eta)
\end{bmatrix}
= -(P\kappa + p).
$$

It has been shown in Theorem 2 in [67] that the real parts of all eigenvalues of $P$ are strictly positive, which guarantees that there exists a positive constant $\lambda_P$ such that

$$
\langle Px, x \rangle \geq \lambda_P \|x\|_2^2 \quad \text{for all} \quad x \in \mathbb{R}^{2d_\rho}. \tag{38}
$$

We also define $\kappa_t = [w_{f,t}^\top, w_{\rho,t}^\top, \eta_t]^\top$. The update of density ratio learning can be rewritten as

$$
\kappa_{t+1} = \kappa_t - \beta_t \zeta(x_t, \kappa_t),
$$

where $\zeta(x_t, \kappa_t) = P_t \kappa_t + p_t$. We also define the population update as $\zeta(\kappa_t) = \mathbb{E}_{D \cdot \pi}[\zeta(x, \kappa_t)] = P\kappa_t + p$. Without loss of generality, we assume that there exists a positive constant $C_\kappa$ such that $\|\kappa^*\|_2 \leq C_\kappa$, where $\kappa^*$ is the global optimum of the density ratio learning defined as

$$
\langle \zeta(\kappa^*), \kappa - \kappa^* \rangle \leq 0, \quad \forall \kappa \in \mathbb{R}^{d_\rho} \times R_\rho \times \mathbb{R}. \tag{39}
$$

The following lemma, often referred to as the "three-points" lemma, characterizes the incremental updating progress of $\kappa_t$ with projection, a proof of which can be found in Lemma 3.1 in [20].

**Lemma 3.** *Consider the update of $w_{f,t}$, $w_{\rho,t}$ and $\eta_t$ in Algorithm 1. For all $\kappa \in \mathbb{R}^M \times R_\rho \times \mathbb{R}$, we have the following holds*

$$
\beta_t \langle \zeta(x_t, \kappa_t), \kappa_{t+1} - \kappa \rangle + \frac{1}{2} \|\kappa_{t+1} - \kappa_t\|_2^2 \leq \frac{1}{2} \|\kappa_t - \kappa\|_2^2 - \frac{1}{2} \|\kappa_{t+1} - \kappa\|_2^2. \tag{40}
$$

Similarly to Lemma 3, we also have the following "three-points lemma" for the iteration of $\theta_t$.

**Lemma 4.** *Consider the update of $\theta_t$ in Algorithm 1. For all $\theta \in R_\theta$, we have the following holds*

$$
-\alpha_t \langle \hat{\rho}(x_t, w_{\rho,t}) g(x_t, \theta_t), \theta_{t+1} - \theta \rangle + \frac{1}{2} \|\theta_{t+1} - \theta_t\|_2^2 \leq \frac{1}{2} \|\theta_t - \theta\|_2^2 - \frac{1}{2} \|\theta_{t+1} - \theta\|_2^2, \tag{41}
$$

*where $\hat{\rho}(x_t, w_{\rho,t})$ is defined in eq. (8).*

The following lemma characterizes the smoothness of $\zeta(\cdot)$.

**Lemma 5.** *For any $\kappa, \kappa' \in \mathbb{R}^{d_\rho} \times R_\rho \times \mathbb{R}$, we have*

$$
\|\zeta(\kappa) - \zeta(\kappa')\|_2 \leq C_P \|\kappa - \kappa'\|_2,
$$

*where $C_P$ is defined in eq. (37).*

*Proof.* Recalling the definition of $\zeta(\kappa) = P\kappa + p$, we can obtain the following

$$
\|\zeta(\kappa) - \zeta(\kappa')\|_2 = \|P(\kappa - \kappa')\|_2 \leq \|P\|_2 \|\kappa - \kappa'\|_2 \leq C_P \|\kappa - \kappa'\|_2,
$$

which completes the proof. $\qquad\square$

Similarly, the following lemma characterizes the smoothness of $g(\theta) = \mathbb{E}_{\mu_\pi}[g(x, \theta)]$.

**Lemma 6.** *In both the forward and backward GVF evaluation settings, for any $\theta, \theta' \in \mathbb{R}^{d_\rho}$, we have*

$$
\|g(\theta) - g(\theta')\|_2 \leq C_g \|\theta - \theta'\|_F,
$$

*where $C_g = (d_g C_\phi C_m + 1) C_\phi$.*

*Proof.* First consider the forward GVF evaluation setting. Recall the definition of $g(\theta)$ and $x = (s, a, s', a')$, we have

$$
g(\theta) = \mathbb{E}_{\mu_\pi}[\phi(s, a)^\top (B(x) + m(x)\phi(s', a')\theta - \phi(s, a)\theta)],
$$

which implies

$$
\begin{aligned}
\|g(\theta) - g(\theta')\|_2 &= \|\mathbb{E}_{\mu_\pi}[\phi(s, a)(m(x)\phi(s', a')(\theta - \theta') + \phi(s, a)(\theta' - \theta))]\|_2 \\
&\leq \mathbb{E}_{\mu_\pi}[(\|\phi(s, a)\|_F \|m(x)\|_F + 1) \|\theta' - \theta\|_2 \|\phi(s, a)\|_F] \\
&\leq C_g \|\theta - \theta'\|_2. \tag{42}
\end{aligned}
$$

Following the steps similar to those in eq. (42), we can also prove that $\|g(\theta) - g(\theta')\|_2 \leq C_g \|\theta - \theta'\|_F$ holds in the backward GVF evaluation setting. $\qquad\square$

The following lemma characterizes the monotonicity of $\zeta(\cdot)$.

**Lemma 7.** *We have the following holds*

$$\langle \zeta(\kappa), \kappa - \kappa^* \rangle \geq \lambda_P \|\kappa - \kappa^*\|_2^2, \quad \forall \kappa \in \mathbb{R}^{d_\rho} \times R_\rho \times \mathbb{R}.$$

*Proof.* Recall that $P$ is strictly positive defined (eq. (38)). We have

$$\begin{aligned}
\langle \zeta(\kappa), \kappa - \kappa^* \rangle &= \langle \zeta(\kappa^*), \kappa - \kappa^* \rangle + \langle \zeta(\kappa) - \zeta(\kappa^*), \kappa - \kappa^* \rangle \\
&\geq \langle \zeta(\kappa) - \zeta(\kappa^*), \kappa - \kappa^* \rangle \\
&= \langle P(\kappa - \kappa^*), \kappa - \kappa^* \rangle \\
&\geq \lambda_P \|\kappa - \kappa^*\|_2^2,
\end{aligned} \tag{43}$$

which completes the proof. $\qquad \square$

The next lemma bounds the per-iteration variance of the update of $\kappa_t$.

**Lemma 8.** *Given a sample $(s_t, a_t, B_t, s_t') \sim \mathcal{D}_d$ and $a_t' \sim \pi(\cdot|s_t')$ and any $\kappa \in \mathbb{R}^{d_\rho} \times R_\rho \times \mathbb{R}$, we have the following holds*

$$\|\zeta(x_t, \kappa) - \zeta(\kappa)\|_2^2 \leq 8C_P^2 \|\kappa - \kappa^*\|_2^2 + 8C_P^2 C_\kappa^2.$$

*Proof.* Recalling the definitions of $\zeta(x_t, \kappa) = P_t \kappa + p_t$ and $\zeta(\kappa) = P\kappa + p$, we can obtain the following

$$\begin{aligned}
\|\zeta(x_t, \kappa) - \zeta(\kappa)\|_2^2 &= \|(P_t - P)\kappa\|_2^2 = 2\|(P_t - P)(\kappa - \kappa^*)\|_2^2 + 2\|(P_t - P)\kappa^*\|_2^2 \\
&\leq 8C_P^2 \|\kappa - \kappa^*\|_2^2 + 8C_P^2 C_\kappa^2.
\end{aligned}$$

$\qquad \square$

The following lemma bounds the norm of the stochastic update $g(x, \theta)$ and the per-iteration variance of GenTD update with density ratio $\rho(s, a)$.

**Lemma 9.** *Given a sample $(s_t, a_t, B_t, s_t') \sim \mathcal{D}$ and $a_t' \sim \pi(\cdot|s_t')$ and any $\theta \in R_\theta$, we have the following holds*

$$\|g(x_t, \theta)\|_2 \leq D_g, \quad \text{and} \quad \mathbb{E}[\|\rho(s_t, a_t)g(x_t, \theta) - g(\theta)\|_2^2] \leq V_g,$$

*where $D_g = d_g C_\phi[C_{\max} + (C_m + 1)D_\theta C_\phi]$ and $V_g = 2\rho_{\max} D_g$.*

*Proof.* We prove the first result as follows,

$$\begin{aligned}
\|g(x_t, \theta)\|_2 &= \left\|\phi(s, a)^\top (B(x) + m(s', a')\phi(s', a')\theta - \phi(s, a)\theta)\right\|_2 \\
&\leq \left\|\phi(s, a)^\top B(x)\right\|_F + \|\phi(s, a)\|_F \left\|m(s', a')\phi(s', a')\theta - \phi(s, a)\theta\right\|_F \\
&\leq d_g C_\phi[C_{\max} + (C_m + 1)D_\theta C_\phi],
\end{aligned}$$

where the last inequality follows from the boundness of the set $R_\theta$. Here we consider $\|\theta\|_2 \leq D_\theta$ for all $\theta \in R_\theta$. The second result can be obtained as follows

$$\|\rho(s_t, a_t)g(x_t, \theta) - g(\theta)\|_2^2 \leq |\rho(s_t, a_t)| \left(\|g(x_t, \theta)\|_2 + \|g(\theta)\|_2\right) \leq 2\rho_{\max} D_g. \tag{44}$$

$\qquad \square$

We next bound the convergence rate of $w_{\rho,t}$.

**Lemma 10.** *Consider $w_{f,t}$, $w_{\rho,t}$ and $\eta_t$ in Algorithm 1. Let stepsize $\beta_t = \frac{2}{\lambda_P(t+t_0+1)}$ where $t_0 = \frac{36C_P^2}{\lambda_P^2}$. For any $\kappa \in \mathbb{R}^{d_\rho} \times R_\rho \times \mathbb{R}$, we have*

$$\mathbb{E}[\|\kappa_T - \kappa^*\|_2^2] \leq \frac{(1 + 16\beta_0^2 C_P^2)(t_0 + 1)^2 \|\kappa_0 - \kappa^*\|_2^2}{(T + t_0 - 1)(T + t_0)} + \frac{64C_P^2 C_\kappa^2}{(T + t_0)\lambda_P^2}.$$

*Proof.* The inner product in eq. (40) can be equivalently written as

$$\langle \zeta(x_t, \kappa_t), \kappa_{t+1} - \kappa \rangle$$
$$= \langle \zeta(\kappa_{t+1}), \kappa_{t+1} - \kappa \rangle + \langle \zeta(\kappa_t) - \zeta(\kappa_{t+1}), \kappa_{t+1} - \kappa \rangle + \langle \zeta(x_t, \kappa_t) - \zeta(\kappa_t), \kappa_t - \kappa \rangle$$
$$+ \langle \zeta(x_t, \kappa_t) - \zeta(\kappa_t), \kappa_{t+1} - \kappa_t \rangle$$
$$\geq \langle \zeta(\kappa_{t+1}), \kappa_{t+1} - \kappa \rangle - \|\zeta(\kappa_t) - \zeta(\kappa_{t+1})\|_2 \|\kappa_{t+1} - \kappa\|_2 + \langle \zeta(x_t, \kappa_t) - \zeta(\kappa_t), \kappa_t - \kappa \rangle$$
$$- \|\zeta(x_t, \kappa_t) - \zeta(\kappa_t)\|_2 \|\kappa_{t+1} - \kappa_t\|_2$$
$$\geq \langle \zeta(\kappa_{t+1}), \kappa_{t+1} - \kappa \rangle - C_P \|\kappa_t - \kappa_{t+1}\|_2 \|\kappa_{t+1} - \kappa\|_2 + \langle \zeta(x_t, \kappa_t) - \zeta(\kappa_t), \kappa_t - \kappa \rangle$$
$$- \|\zeta(x_t, \kappa_t) - \zeta(\kappa_t)\|_2 \|\kappa_{t+1} - \kappa_t\|_2, \tag{45}$$

where the last inequality follows from Lemma 5. Substituting eq. (45) into eq. (40), we obtain

$$\frac{1}{2} \|\kappa_t - \kappa\|_2^2 - \frac{1}{2} \|\kappa_{t+1} - \kappa\|_2^2$$
$$\geq \beta_t \langle \zeta(\kappa_{t+1}), \kappa_{t+1} - \kappa \rangle - \beta_t C_P \|\kappa_t - \kappa_{t+1}\|_2 \|\kappa_{t+1} - \kappa\|_2 + \beta_t \langle \zeta(x_t, \kappa_t) - \zeta(\kappa_t), \kappa_t - \kappa \rangle$$
$$- \beta_t \|\zeta(x_t, \kappa_t) - \zeta(\kappa_t)\|_2 \|\kappa_{t+1} - \kappa_t\|_2 + \frac{1}{2} \|\kappa_{t+1} - \kappa_t\|_2^2. \tag{46}$$

Note that we have the following holds

$$\frac{1}{2} \|\kappa_{t+1} - \kappa_t\|_2^2 - \beta_t C_P \|\kappa_t - \kappa_{t+1}\|_2 \|\kappa_{t+1} - \kappa\|_2 - \beta_t \|\zeta(x_t, \kappa_t) - \zeta(\kappa_t)\|_2 \|\kappa_{t+1} - \kappa_t\|_2$$
$$= \frac{1}{4} \|\kappa_{t+1} - \kappa_t\|_2^2 - \beta_t C_P \|\kappa_t - \kappa_{t+1}\|_2 \|\kappa_{t+1} - \kappa\|_2 + \frac{1}{4} \|\kappa_{t+1} - \kappa_t\|_2^2$$
$$- \beta_t \|\zeta(x_t, \kappa_t) - \zeta(\kappa_t)\|_2 \|\kappa_{t+1} - \kappa_t\|_2$$
$$\geq -\beta_t^2 C_P^2 \|\kappa_{t+1} - \kappa\|_2^2 - \beta_t^2 \|\zeta(x_t, \kappa_t) - \zeta(\kappa_t)\|_2^2. \tag{47}$$

Substituting eq. (47) in eq. (46) yields

$$\frac{1}{2} \|\kappa_t - \kappa\|_2^2 - \frac{1}{2} \|\kappa_{t+1} - \kappa\|_2^2$$
$$\geq \beta_t \langle \zeta(\kappa_{t+1}), \kappa_{t+1} - \kappa \rangle + \beta_t \langle \zeta(x_t, \kappa_t) - \zeta(\kappa_t), \kappa_t - \kappa \rangle - \beta_t^2 C_P^2 \|\kappa_{t+1} - \kappa\|_2^2$$
$$- \beta_t^2 \|\zeta(x_t, \kappa_t) - \zeta(\kappa_t)\|_2^2. \tag{48}$$

Rearranging eq. (48) and letting $\kappa = \kappa^*$ yield

$$\|\kappa_t - \kappa^*\|_2^2 + 2\beta_t^2 \|\zeta(x_t, \kappa_t) - \zeta(\kappa_t)\|_2^2$$
$$\geq (1 - 2\beta_t^2 C_P^2) \|\kappa_{t+1} - \kappa^*\|_2^2 + 2\beta_t \langle \zeta(\kappa_{t+1}), \kappa_{t+1} - \kappa^* \rangle + 2\beta_t \langle \zeta(x_t, \kappa_t) - \zeta(\kappa_t), \kappa_t - \kappa^* \rangle$$
$$\geq (1 + 2\beta_t \lambda_P - 2\beta_t^2 C_P^2) \|\kappa_{t+1} - \kappa^*\|_2^2 + 2\beta_t \langle \zeta(x_t, \kappa_t) - \zeta(\kappa_t), \kappa_t - \kappa^* \rangle, \tag{49}$$

where the last inequality follows from $\langle \zeta(\kappa_{t+1}), \kappa_{t+1} - \kappa^* \rangle \leq \lambda_P \|\kappa_{t+1} - \kappa^*\|_2^2$. Taking expectation on both sides of eq. (49), and noting that $\mathbb{E}[\langle \zeta(x_t, \kappa_t) - \zeta(\kappa_t), \kappa_t - \kappa^* \rangle | \mathcal{F}_t] = 0$, we obtain

$$(1 + 2\beta_t \lambda_P - 2\beta_t^2 C_P^2)\mathbb{E}[\|\kappa_{t+1} - \kappa^*\|_2^2]$$
$$\leq \mathbb{E}[\|\kappa_t - \kappa^*\|_2^2] + 2\beta_t^2 \mathbb{E}[\|\zeta(x_t, \kappa_t) - \zeta(\kappa_t)\|_2^2]$$
$$\leq (1 + 16\beta_t^2 C_P^2)\mathbb{E}[\|\kappa_t - \kappa^*\|_2^2] + 16\beta_t^2 C_P^2 C_\kappa^2. \tag{50}$$

Multiplying both sides of eq. (50) with $I_t$ and summing over $t = 0, \cdots, T-1$ yield

$$\sum_{t=0}^{T-1} a_t \mathbb{E}[\|\kappa_{t+1} - \kappa^*\|_2^2] \leq \sum_{t=0}^{T-1} b_t \mathbb{E}[\|\kappa_t - \kappa^*\|_2^2] + c, \tag{51}$$

where

$$a_t = (1 + 2\beta_t \lambda_P - 2\beta_t^2 C_P^2)I_t,$$
$$b_t = (1 + 16\beta_t^2 C_P^2)I_t,$$
$$c = 16 C_P^2 C_\kappa^2 \sum_{t=0}^{T-1} \beta_t^2 I_t.$$

We further let

$$I_t = (t + t_0)(t + t_0 + 1),$$

$$\beta_t = \frac{2}{\lambda_P(t + t_0 - 1)},$$

$$t_0 = \frac{36C_P^2}{\lambda_P^2} + 1.$$

We can obtain the following

$$
\begin{aligned}
a_t - b_{t+1} &= (1 + 2\beta_t\lambda_P - 2\beta_t^2 C_P^2)I_t - (1 + 16\beta_{t+1}^2 C_P^2)s_{t+1} \\
&\geq (1 + 2\beta_t\lambda_P)I_t - (1 + 2\beta_t^2 + 16\beta_{t+1}^2 C_P^2)s_{t+1} \\
&\geq (1 + 2\beta_t\lambda_P)I_t - (1 + 18\beta_t^2 C_P^2)s_{t+1} \\
&\geq (1 + 2\beta_t\lambda_P)I_t - (1 + \beta_t\lambda_P)s_{t+1} \\
&\geq (t + t_0 + 1)\frac{(t + t_0 + 3)(t + t_0) - (t + t_0 + 2)^2}{t + t_0 - 1} \\
&\geq 0.
\end{aligned}
$$

Substituting the above results into eq. (51) yields

$$a_{T-1}t\mathbb{E}[\|\kappa_T - \kappa^*\|_2^2] \leq b_0 \|\kappa_0 - \kappa^*\|_2^2 + c,$$

which implies

$$
\begin{aligned}
\mathbb{E}[\|\kappa_T - \kappa^*\|_2^2] &\leq \frac{b_0 \|\kappa_0 - \kappa^*\|_2^2}{A_{T-1}} + \frac{c}{a_{T-1}} \\
&= \frac{(1 + 16\beta_0^2 C_P^2)s_0 \|\kappa_0 - \kappa^*\|_2^2}{(1 + 2\beta_{T-1}\lambda_P - 2\beta_{T-1}^2 C_P^2)s_{T-1}} + \frac{16C_P^2 C_\kappa^2 \sum_{t=0}^{T-1} \beta_t^2 I_t}{(1 + 2\beta_{T-1}\lambda_P - 2\beta_{T-1}^2 C_P^2)s_{T-1}} \\
&\leq \frac{(1 + 16\beta_0^2 C_P^2)(t_0 + 1)^2 \|\kappa_0 - \kappa^*\|_2^2}{(T + t_0 - 1)(T + t_0)} + \frac{64C_P^2 C_\kappa^2}{(T + t_0)\lambda_P^2},
\end{aligned}
$$

which completes the proof. $\qquad\square$

Note that Lemma 10 implies that there exists a positive number $D_\rho$ such that

$$\mathbb{E}[\|w_{\rho,t} - w_\rho^*\|_2^2] \leq \frac{D_\rho}{t + t_0}. \tag{52}$$

### F.2 Proof of Theorem 1

Consider the inner product term in eq. (41). We have

$$
\begin{aligned}
&-\langle \hat{\rho}(x_t, w_{\rho,t})g(x_t, \theta_t), \theta_{t+1} - \theta \rangle \\
&= -\langle g(\theta_{t+1}), \theta_{t+1} - \theta \rangle - \langle g(\theta_t - g(\theta_{t+1}), \theta_{t+1} - \theta \rangle \\
&\quad - \langle \rho(x_t)g(x_t, \theta_t) - g(\theta_t), \theta_{t+1} - \theta_t \rangle - \langle \rho(x_t)g(x_t, \theta_t) - g(\theta_t), \theta_t - \theta \rangle \\
&\quad - \langle (\hat{\rho}(x_t, w_\rho^*) - \rho(x_t))g(x_t, \theta_t), \theta_{t+1} - \theta \rangle \\
&\quad - \langle (\hat{\rho}(x_t, w_{\rho,t}) - \hat{\rho}(x_t, w_\rho^*))g(x_t, \theta_t), \theta_{t+1}^\top - \theta \rangle \\
&\geq -\langle g(\theta_{t+1}), \theta_{t+1}^\top - \theta \rangle - C_g \|\theta_t - \theta_{t+1}\|_2 \|\theta_{t+1} - \theta\|_2 \\
&\quad - \|\rho(x_t)g(x_t, \theta_t) - g(\theta_t)\|_2 \|\theta_{t+1} - \theta_t\|_2 - \langle \rho(x_t)g(x_t, \theta_t) - g(\theta_t), \theta_t^\top - \theta \rangle \\
&\quad - |\hat{\rho}(x_t, w_\rho^*) - \rho(x_t)| \|g(x_t, \theta_t)\|_2 \|\theta_{t+1} - \theta\|_2 \\
&\quad - |\hat{\rho}(x_t, w_{\rho,t}) - \hat{\rho}(x_t, w_\rho^*)| \|g(x_t, \theta_t)\|_2 \|\theta_{t+1} - \theta\|_2, \tag{53}
\end{aligned}
$$

where the last inequality follows from Lemma 6. Substituting eq. (53) into eq. (41) yields

$$\frac{1}{2} \|\theta_t - \theta\|_2^2 - \frac{1}{2} \|\theta_{t+1} - \theta\|_2^2$$

$$\geq -\alpha_t \langle g(\theta_{t+1}), \theta_{t+1}^\top - \theta \rangle - \alpha_t C_g \|\theta_t - \theta_{t+1}\|_2 \|\theta_{t+1} - \theta\|_2$$
$$- \alpha_t \|\rho(x_t)g(x_t, \theta_t) - g(\theta_t)\|_2 \|\theta_{t+1} - \theta_t\|_2 - \alpha_t \langle \rho(x_t)g(x_t, \theta_t) - g(\theta_t), \theta_t^\top - \theta^\top \rangle$$
$$- \alpha_t \left| \hat\rho(x_t, w_\rho^*) - \rho(x_t) \right| \|g(x_t, \theta_t)\|_2 \|\theta_{t+1} - \theta\|_2$$
$$- \alpha_t \left| \hat\rho(x_t, w_{\rho,t}) - \hat\rho(x_t, w_\rho^*) \right| \|g(x_t, \theta_t)\|_2 \|\theta_{t+1} - \theta\|_2 + \frac{1}{2} \|\theta_{t+1} - \theta_t\|_2^2 . \tag{54}$$

We have the following holds

$$\frac{1}{2} \|\theta_{t+1} - \theta_t\|_2^2 - \alpha_t C_g \|\theta_t - \theta_{t+1}\|_2 \|\theta_{t+1} - \theta\|_2 - \alpha_t \|\rho(x_t)g(x_t, \theta_t) - g(\theta_t)\|_2 \|\theta_{t+1} - \theta_t\|_2$$
$$= \frac{1}{4} \|\theta_{t+1} - \theta_t\|_2^2 - \alpha_t C_g \|\theta_t - \theta_{t+1}\|_2 \|\theta_{t+1} - \theta\|_2 + \frac{1}{4} \|\theta_{t+1} - \theta_t\|_2^2$$
$$\quad - \alpha_t \|\rho(x_t)g(x_t, \theta_t) - g(\theta_t)\|_2 \|\theta_{t+1} - \theta_t\|_2$$
$$\geq -\alpha_t^2 C_g^2 \|\theta_{t+1} - \theta\|_2^2 - \alpha_t^2 \|\rho(x_t)g(x_t, \theta_t) - g(\theta_t)\|_2^2 , \tag{55}$$

which implies

$$\frac{1}{2} \|\theta_t - \theta\|_2^2 - \frac{1}{2} \|\theta_{t+1} - \theta\|_2^2$$
$$\geq -\alpha_t \langle g(\theta_{t+1}), \theta_{t+1} - \theta \rangle - \alpha_t^2 C_g^2 \|\theta_{t+1} - \theta\|_2^2 - \alpha_t^2 \|\rho(x_t)g(x_t, \theta_t) - g(\theta_t)\|_2^2$$
$$\quad - \alpha_t \langle \rho(x_t)g(x_t, \theta_t) - g(\theta_t), \theta_t - \theta \rangle - \alpha_t D_g \left| \hat\rho(x_t, w_\rho^*) - \rho(x_t) \right| \|\theta_{t+1} - \theta\|_2$$
$$\quad - \alpha_t D_g \left| \hat\rho(x_t, w_{\rho,t}) - \hat\rho(x_t, w_\rho^*) \right| \|\theta_{t+1} - \theta\|_2 , \tag{56}$$

where we use the fact that $\|g(x_t, \theta_t)\|_2 \leq D_g$ in Lemma 9. Rearranging eq. (56) and letting $\theta = \theta^*$ yield

$$\|\theta_t - \theta^*\|_2^2 + 2\alpha_t^2 \|\rho(x_t)g(x_t, \theta_t) - g(\theta_t)\|_2^2$$
$$\geq \|\theta_{t+1} - \theta^*\|_2^2 - 2\alpha_t \langle g(\theta_{t+1}), \theta_{t+1} - \theta^* \rangle - 2\alpha_t^2 C_g^2 \|\theta_{t+1} - \theta^*\|_2^2$$
$$\quad - 2\alpha_t \langle \rho(x_t)g(x_t, \theta_t) - g(\theta_t), \theta_t - \theta^* \rangle - 2\alpha_t D_g \left| \hat\rho(x_t, w_\rho^*) - \rho(x_t) \right| \|\theta_{t+1} - \theta^*\|_2$$
$$\quad - 2\alpha_t D_g \left| \hat\rho(x_t, w_{\rho,t}) - \hat\rho(x_t, w_\rho^*) \right| \|\theta_{t+1} - \theta\|_2$$
$$\geq (1 + 2\alpha_t \lambda_g - 2\alpha_t^2 C_g^2) \|\theta_{t+1} - \theta^*\|_2^2 - 2\alpha_t \langle \rho(x_t)g(x_t, \theta_t) - g(\theta_t), \theta_t - \theta^* \rangle$$
$$\quad - 2\alpha_t D_g \left| \hat\rho(x_t, w_\rho^*) - \rho(x_t) \right| \|\theta_{t+1} - \theta^*\|_2$$
$$\quad - 2\alpha_t D_g \left| \hat\rho(x_t, w_{\rho,t}) - \hat\rho(x_t, w_\rho^*) \right| \|\theta_{t+1} - \theta\|_2$$
$$\geq (1 + 2\alpha_t \lambda_g - 2\alpha_t^2 C_g^2) \|\theta_{t+1} - \theta^*\|_2^2 - 2\alpha_t \langle \rho(x_t)g(x_t, \theta_t) - g(\theta_t), \theta_t - \theta^* \rangle$$
$$\quad - \frac{1}{2}\alpha_t \lambda_g \|\theta_{t+1} - \theta^*\|_2^2 - \frac{2\alpha_t D_g^2}{\lambda_g} \left| \hat\rho(x_t, w_\rho^*) - \rho(x_t) \right|^2$$
$$\quad - \frac{1}{2}\alpha_t \lambda_g \|\theta_{t+1} - \theta^*\|_2^2 - \frac{2\alpha_t D_g^2}{\lambda_g} \left| \hat\rho(x_t, w_{\rho,t}) - \hat\rho(x_t, w_\rho^*) \right|^2$$
$$= (1 + \alpha_t \lambda_g - 2\alpha_t^2 C_g^2) \|\theta_{t+1} - \theta^*\|_2^2 - 2\alpha_t \langle \rho(x_t)g(x_t, \theta_t) - g(\theta_t), \theta_t - \theta^* \rangle$$
$$\quad - \frac{2\alpha_t D_g^2}{\lambda_g} \left| \hat\rho(x_t, w_\rho^*) - \rho(x_t) \right|^2 - \frac{2\alpha_t D_g^2}{\lambda_g} \left| \hat\rho(x_t, w_{\rho,t}) - \hat\rho(x_t, w_\rho^*) \right|^2 , \tag{57}$$

where the first inequality follows from Lemma 4, and the third inequality follows from Young's inequality. Taking expectation on both sides of eq. (57) yields

$$(1 + \alpha_t \lambda_g - 2\alpha_t^2 C_g^2)\mathbb{E}[\|\theta_{t+1} - \theta^*\|_2^2]$$
$$\leq \mathbb{E}[\|\theta_t - \theta^*\|_2^2] + 2\alpha_t^2 \mathbb{E}[\|\rho(x_t)g(x_t, \theta_t) - g(\theta_t)\|_2^2] + \frac{2\alpha_t D_g^2}{\lambda_g} \mathbb{E}\left[ \left| \hat\rho(x_t, w_\rho^*) - \rho(x_t) \right|^2 \right]$$
$$\quad + \frac{2\alpha_t D_g^2 C_\psi^2}{\lambda_g} \mathbb{E}\left[ \left| \hat\rho(x_t, w_{\rho,t}) - \hat\rho(x_t, w_\rho^*) \right|^2 \right]$$

$$\leq \mathbb{E}[\|\theta_t - \theta^*\|_2^2] + 2V_g\alpha_t^2 + \frac{2\alpha_t D_g^2 C_\psi^2}{\lambda_g}\mathbb{E}\left[\|w_{\rho,t} - w_\rho^*\|_2^2\right] + \frac{2D_g^2\alpha_t\varepsilon_\rho}{\lambda_g}, \quad (58)$$

where the last inequality follows from Lemma 9.

Substituting eq. (52) into eq. (58) yields

$$(1 + \alpha_t\lambda_g - 2\alpha_t^2 C_g^2)\mathbb{E}[\|\theta_{t+1} - \theta^*\|_2^2]$$

$$\leq \mathbb{E}[\|\theta_t - \theta^*\|_2^2] + 2V_g\alpha_t^2 + \frac{2\alpha_t D_g^2 D_\rho C_\psi^2}{\lambda_g(t + t_0)} + \frac{2D_g^2\alpha_t\varepsilon_\rho}{\lambda_g}. \quad (59)$$

Multiplying both sides of eq. (59) with $r_t$ and summing over $t = 0, \cdots, T - 1$ yield

$$\sum_{t=0}^{T-1} a_t'\mathbb{E}[\|\theta_{t+1} - \theta^*\|_2^2]$$

$$\leq \sum_{t=0}^{T-1} r_t\mathbb{E}[\|\theta_t - \theta^*\|_2^2] + 2V_g\sum_{t=0}^{T-1} r_t\alpha_t^2 + \frac{2D_g^2 D_\rho C_\psi^2}{\lambda_g(t + t_0)}\sum_{t=0}^{T-1} r_t\alpha_t + \frac{2D_g^2\varepsilon_\rho}{\lambda_g}\sum_{t=0}^{T-1} r_t\alpha_t, \quad (60)$$

where

$$a_t' = (1 + \alpha_t\lambda_g - 2\alpha_t^2 C_g^2)r_t.$$

Now we let

$$r_t = (t + t_1)(t + t_1 + 1),$$

$$\alpha_t = \frac{4}{\lambda_g(t + t_1 - 1)},$$

$$t_1 = \frac{16C_g^2}{\lambda_g^2} + 1.$$

We can obtain the following

$$a_t' - r_{t+1} = (1 + \alpha_t\lambda_g - 2\alpha_t^2 C_g^2)r_t - r_{t+1}$$

$$\geq (1 + \alpha_t\lambda_g)r_t - (1 + 2\alpha_t^2 C_g^2)r_{t+1}$$

$$\geq (1 + \alpha_t\lambda_g)r_t - \left(1 + \frac{1}{2}\alpha_t\lambda_g\right)r_{t+1}$$

$$\geq (t + t_1 + 1)\frac{(t + t_1)(t + t_1 + 3) - (t + t_1 + 2)^2}{t + t_1 + 1}$$

$$\geq 0,$$

where the second inequality follows from the fact that $\alpha_t \leq \frac{\lambda_g}{4C_g^2}$.

Substituting the above result to eq. (60) yields

$$a_{T-1}'\mathbb{E}[\|\theta_T - \theta^*\|_2^2] \leq r_0\|\theta_0 - \theta^*\|_2^2 + 2V_g\sum_{t=0}^{T-1} r_t\alpha_t^2 + \frac{2D_g^2 D_\rho C_\psi^2}{\lambda_g(t + t_0)}\sum_{t=0}^{T-1} r_t\alpha_t + \frac{2D_g^2\varepsilon_\rho}{\lambda_g}\sum_{t=0}^{T-1} r_t\alpha_t.$$

The above inequality implies the following convergence rate

$$\mathbb{E}[\|\theta_T - \theta^*\|_2^2] \leq \frac{r_0\|\theta_0 - \theta^*\|_2^2}{(T + t_1 - 1))(T + t_1)} + \frac{128V_g}{\lambda_g^2(T + t_1)} + \frac{64D_g^2 C_g^2 C_\psi^2\lambda_P^2}{9C_P^2\lambda_g^3(T + t_1)} + \frac{16D_g^2}{\lambda_g^2}\varepsilon_\rho,$$

which completes the proof.

## G  Proof of Theorem 2

Following the similar argument similar to that in Lemma 4.2 in [3] and Theorem 1 in [55], we can prove that $\Phi\theta^*$ is the fixed point of the composite operator $\Gamma_{\Phi,\mu_\pi}\bar{\mathcal{T}}_\pi$. We then proceed as follows

$$\|\Phi\theta^* - G_\pi\|_{\mu_\pi,\alpha} = \left\|\Gamma_{\Phi,\mu_\pi}\bar{\mathcal{T}}_\pi\Phi\theta^* - \Gamma_{\Phi,\mu_\pi}G_\pi + \Gamma_{\Phi,\mu_\pi}G_\pi - G_\pi\right\|_{\mu_\pi,\alpha}$$

$$\leq \left\|\Gamma_{\Phi,\mu_\pi} \bar{\mathcal{T}}_\pi \Phi\theta^* - \Gamma_{\Phi,\mu_\pi} G_\pi\right\|_{\mu_\pi,\alpha} + \left\|\Gamma_{\Phi,\mu_\pi} G_\pi - G_\pi\right\|_{\mu_\pi,\alpha}$$

$$= \left\|\Gamma_{\Phi,\mu_\pi} \bar{\mathcal{T}}_\pi \Phi\theta^* - \Gamma_{\Phi,\mu_\pi} \bar{\mathcal{T}}_\pi G_\pi\right\|_{\mu_\pi,\alpha} + \left\|\Gamma_{\Phi,\mu_\pi} G_\pi - G_\pi\right\|_{\mu_\pi,\alpha}$$

$$\leq \left\|\Gamma_{\Phi,\mu_\pi}[\bar{\mathcal{T}}_\pi \Phi\theta^* - \bar{\mathcal{T}}_\pi G_\pi]\right\|_{\mu_\pi,\alpha} + \left\|\Gamma_{\Phi,\mu_\pi} G_\pi - G_\pi\right\|_{\mu_\pi,\alpha}$$

$$\leq \left\|\bar{\mathcal{T}}_\pi \Phi\theta^* - \bar{\mathcal{T}}_\pi G_\pi\right\|_{\mu_\pi,\alpha} + \left\|\Gamma_{\Phi,\mu_\pi} G_\pi - G_\pi\right\|_{\mu_\pi,\alpha}$$

$$\leq \gamma_G \left\|\Phi\theta^* - G_\pi\right\|_{\mu_\pi,\alpha} + \left\|\Gamma_{\Phi,\mu_\pi} G_\pi - G_\pi\right\|_{\mu_\pi,\alpha}, \tag{61}$$

where the first equality follows from the fact that $\Gamma_{\Phi,\mu_\pi}\bar{\mathcal{T}}_\pi \Phi\theta^* = \Phi\theta^*$, the second equality follows from the fact that $\bar{\mathcal{T}}_\pi G_\pi = G_\pi$, the third inequality follows from the non-expansive property of the projection operator $\Gamma_{\Phi,\mu_\pi}$, and the last inequality follows from Proposition 1. Equation (61) implies the following result

$$\left\|\Phi\theta^* - G_\pi\right\|_{\mu_\pi,\alpha} \leq \frac{1}{1-\gamma_G} \left\|\Gamma_{\Phi,\mu_\pi} G_\pi - G_\pi\right\|_{\mu_\pi,\alpha}.$$

## H    Extension to Case $\gamma_{\max} = 1$

As shown in Proposition 1, the operator $\bar{\mathcal{T}}_{G,\pi}$ is not necessarily a contraction when $\gamma_{\max} = 1$. The uniqueness of $G_\pi$ and $\hat{G}_\pi$ is not guaranteed in this case. We next consider the following assumption for the base matrix $\Phi_i$, which can yield a desired property as we show below. Such an assumption has also been considered in the average reward MDP setting [56].

**Assumption 6** (Non-constant Parameterization). *For all $i = 1, \cdots, k$, we have $\Phi_i\theta_i \neq c\mathbf{1}$ for any $\theta_i \in \mathbb{R}^{d_i}$ and $c \in \mathbb{R}/0$.*

Despite the non-contraction nature of $\bar{\mathcal{T}}_{G,\pi}$, if the base function $\Phi_i$ satisfies Assumption 6, we can show that the monotonicity condition of $g(\theta)$ in Proposition 2 still holds with a positive constant $\lambda_G$. As a result, the convergence bound in eq. (12) of theorem 1 is directly applicable to this setting with the corresponding value of $\lambda_G$. We can then further establish a result similar to Theorem 2 for the case with $\gamma_{\max} = 1$ under Assumption 6.

We first extend Proposition 2 and Theorem 2 to the case in which $\gamma_{\max} = 1$. Without loss of generality, we consider $\gamma_i = 1$ for all $i = 1, \cdots, k$.

**Forward GVF.** We first verify Proposition 2. In this setting, we can still obtain the same result for $G$ as in eq. (30), but with $A_i = [\Phi_i^\top \bar{U}_\pi(P_\pi - I)\Phi_i] \otimes I_{d_i}$, where $\bar{U}_\pi = \text{diag}(\mu_\pi)$. As shown in Lemma 7 in [56], the matrix $[\Phi_i^\top U_{\pi,i}(P_\pi - I)\Phi_i]$ is Hurwitz when the base matrix $\Phi_i$ satisfies Assumption 6. Following the steps similar to those in eq. (31) - (34), we can conclude that the matrix $G$ is also Hurwitz, which completes the proof.

We then verify Theorem 2. We proceed as follows,

$$\|\Phi\theta^* - G_\pi\|_{\mu_\pi,\alpha}$$

$$= \|\Gamma_{\Phi,\mu_\pi} \mathcal{T}_\pi \Phi\theta^* - \mathcal{T}_\pi G_\pi\|_{\mu_\pi,\alpha}$$

$$\leq \|\Gamma_{\Phi,\mu_\pi} \mathcal{T}_\pi \Phi\theta^* - \Gamma_{\Phi,\mu_\pi} \mathcal{T}_\pi G_\pi\|_{\mu_\pi,\alpha} + \|\Gamma_{\Phi,\mu_\pi} \mathcal{T}_\pi G_\pi - \mathcal{T}_\pi G_\pi\|_{U_\pi}$$

$$\leq \|\Gamma_{\Phi,\mu_\pi} M_\pi(\Phi\theta^* - G_\pi)\|_{U_\pi} + \|\Gamma_{\Phi,\mu_\pi} \mathcal{T}_\pi G_\pi - \mathcal{T}_\pi G_\pi\|_{\mu_\pi,\alpha}$$

$$\leq \|M_\pi(\Phi\theta^* - G_\pi)\|_{U_\pi} + \|\Gamma_{\Phi,\mu_\pi} \mathcal{T}_\pi G_\pi - \mathcal{T}_\pi G_\pi\|_{\mu_\pi,\alpha}$$

$$\leq C_\zeta \|\Phi\theta^* - G_\pi\|_{U_\pi} + \|\Gamma_{\Phi,\mu_\pi} \mathcal{T}_\pi G_\pi - \mathcal{T}_\pi G_\pi\|_{\mu_\pi,\alpha}, \tag{62}$$

where the last inequality in eq. (62) can be obtained as follows. Following the steps similar to those in eq. (31)-(34), we can conclude that $\text{eig}(M_\pi) = \text{eig}(P_\pi)$. For an ergodic MDP, we have $\max[\text{eig}(P_\pi)] = \max[\text{eig}(P_\pi^\top)] = 1$. Let $i = \arg\max_j[\text{eig}(P_\pi)_j]$. We then have $\max_{j \neq i} \text{eig}(P_\pi)_j < 1$. Let $G_\pi$ be the fixed point of $\mathcal{T}_\pi$ that is perpendicular to $[c_1\mathbf{1}_{d_1}, \cdots, c_k\mathbf{1}_{d_k}]$, where $c_1, \cdots, c_k$ could be any constant. The vector $\Phi\theta^* - G_\pi$ is perpendicular to the space spanned by the eigenvectors of $M_\pi$ associated with the eigenvalue 1. Thus, there exists a positive constant $C_\zeta < 1$ such that $\|M_\pi(\Phi\theta^* - G_\pi)\|_{U_\pi} \leq C_\zeta \|\Phi\theta^* - G_\pi\|_{U_\pi}$, which yields the following results

$$\|\Phi\theta^* - G_\pi\|_{\mu_\pi,\alpha} \leq \frac{1}{1-C_\zeta} \|\Gamma_{\Phi,\mu_\pi} \mathcal{T}_\pi G_\pi - \mathcal{T}_\pi G_\pi\|_{\mu_\pi,\alpha}. \tag{63}$$

**Bakcward GVF.** To verify Proposition 2, we can obtain the same result for $G$ as in eq. (30) with $A_i = [\Phi_i^\top (\mathsf{P}_\pi^\top - I) \bar{U}_\pi \Phi_i] \otimes \mathsf{I}_{d_i}$. Define $\bar{A}_i = \Phi_i^\top (\mathsf{P}_\pi^\top - I) U_\pi \Phi_i$. We next show that $\bar{A}_i$ is Hurwitz. Note that $\bar{A}_i = \mathbb{E}_{\mu_\pi}[\phi'(\phi - \phi')]$. Let $z$ be a non-constant function on the state-action space. Then we have

$$
\begin{aligned}
0 &< \frac{1}{2} \mathbb{E}_{\mu_\pi}[(z(s,a) - z(s',a'))^2] \\
&= \mathbb{E}_{\mu_\pi}[z(s,a)^2] - \mathbb{E}[z(s,a)z(s',a')] \\
&= z^\top \bar{U}_\pi z - z^\top \mathsf{P}_\pi \bar{U}_\pi z \\
&= z^\top (I - \mathsf{P}_\pi) \bar{U}_\pi z.
\end{aligned}
\tag{64}
$$

For a vector $v \in \mathbb{R}^{K_i}$, we have

$$
v^\top \bar{A}_i v = v \Phi_i^\top (\mathsf{P}_\pi^\top - I) U_\pi \Phi_i v.
\tag{65}
$$

Since $\Phi v$ is a non-constant function, eq. (64) and eq. (65) together imply that

$$
v^\top \bar{A}_i v < 0 \quad \text{for all} \quad v \in \mathbb{R}^{K_i}.
$$

Thus, the matrix $\bar{A}_i$ is Hurwitz, which further implies that $A_i$ is also Hurwitz. Following the steps similar to those in eq. (31) - (34), we can conclude that the matrix $G$ is also Hurwitz, which completes the proof.

We then verify Theorem 2. We proceed as follows,

$$
\begin{aligned}
&\|\Phi\theta^* - G_\pi\|_{\mu_\pi,\alpha} \\
&= \|\Gamma_{\Phi,\mu_\pi} \mathcal{T}_\pi \Phi\theta^* - \mathcal{T}_\pi G_\pi\|_{\mu_\pi,\alpha} \\
&\leq \|\Gamma_{\Phi,\mu_\pi} \mathcal{T}_\pi \Phi\theta^* - \Gamma_{\Phi,\mu_\pi} \mathcal{T}_\pi G_\pi\|_{\mu_\pi,\alpha} + \|\Gamma_{\Phi,\mu_\pi} \mathcal{T}_\pi G_\pi - \mathcal{T}_\pi G_\pi\|_{U_\pi} \\
&\leq \left\|\Gamma_{\Phi,\mu_\pi} \hat{M}_\pi (\Phi\theta^* - G_\pi)\right\|_{U_\pi} + \|\Gamma_{\Phi,\mu_\pi} \mathcal{T}_\pi G_\pi - \mathcal{T}_\pi G_\pi\|_{\mu_\pi,\alpha} \\
&\leq \left\|\hat{M}_\pi (\Phi\theta^* - G_\pi)\right\|_{U_\pi} + \|\Gamma_{\Phi,\mu_\pi} \mathcal{T}_\pi G_\pi - \mathcal{T}_\pi G_\pi\|_{\mu_\pi,\alpha} \\
&\leq C_\zeta \|\Phi\theta^* - G_\pi\|_{U_\pi} + \|\Gamma_{\Phi,\mu_\pi} \mathcal{T}_\pi G_\pi - \mathcal{T}_\pi G_\pi\|_{\mu_\pi,\alpha},
\end{aligned}
\tag{66}
$$

where the last inequality in eq. (66) can be obtained as follows. Using Theorem 1.3.22 in [13], we have

$$
\text{eig}(U_{\pi,i}^{-1}[\mathsf{P}_{\pi,i}^\top \otimes \mathsf{I}_{d_i}]U_{\pi,i}) = \text{eig}([\mathsf{P}_{\pi,i}^\top \otimes \mathsf{I}_{d_i}]U_{\pi,i}U_{\pi,i}^{-1}) = \text{eig}([\mathsf{P}_\pi^\top \otimes \mathsf{I}_{d_i}]) = \text{eig}(\mathsf{P}_\pi^\top) = \text{eig}(\mathsf{P}_\pi).
$$

Following the steps similar to those in eq. (31)-(34), we can conclude that $\text{eig}(\hat{M}_\pi) = \text{eig}(\mathsf{P}_\pi)$. Following the steps similar to those for obtaining eq. (63). We have

$$
\|\Phi\theta^* - G_\pi\|_{\mu_\pi,\alpha} \leq \frac{1}{1 - C_\zeta} \|\Gamma_{\Phi,\mu_\pi} \mathcal{T}_\pi G_\pi - \mathcal{T}_\pi G_\pi\|_{\mu_\pi,\alpha},
$$

where $0 < C_\zeta < 1$, which completes the proof.

# I   Proof of Theorem 3

We first define the matrix $B$ in the following way:

- Forward GVF: $B = \mathbb{E}_{\mathcal{D}\cdot\pi}[[\phi(s',a') \otimes \mathsf{I}_d]m(x)[\phi(s,a) \otimes \mathsf{I}_d]]$
- Backward GVF: $B = \mathbb{E}_{\mathcal{D}\cdot\pi}[[\phi(s,a) \otimes \mathsf{I}_d]m(x)[\phi(s',a') \otimes \mathsf{I}_d]]$.

We further define the following stochastic matrices in both the forward and backward GVF evaluation settings. Recall that $(s_t, a_t) \sim D(\cdot)$, $s'_t \sim \mathsf{P}(\cdot|s_t, a_t)$ and $a'_t \sim \pi(\cdot|s'_t$.

- Forward GVF:

$$
\begin{aligned}
A_t &= [\phi(s_t, a_t) \otimes \mathsf{I}_d](m(x_t)[\phi(s'_t, a'_t) \otimes \mathsf{I}_d]^\top - [\phi(s_t, a_t) \otimes \mathsf{I}_d]^\top), \\
B_t &= [\phi(s'_t, a'_t) \otimes \mathsf{I}_d]m(x_t)[\phi(s_t, a_t) \otimes \mathsf{I}_d], \\
C_t &= (\phi(s_t, a_t)\phi(s_t, a_t)^\top) \otimes \mathsf{I}_d, \\
b_t &= [\phi(s_t, a_t) \otimes \mathsf{I}_d]C(x_t).
\end{aligned}
\tag{67}
$$

- Backward GVF:

$$A_t = [\phi(s'_t, a'_t) \otimes I_d](m(x_t)[\phi(s_t, a_t) \otimes I_d]^\top - [\phi(s'_t, a'_t) \otimes I_d]^\top),$$
$$B_t = [\phi(s_t, a_t) \otimes I_d]m(x_t)[\phi(s'_t, a'_t) \otimes I_d],$$
$$C_t = (\phi(s'_t, a'_t)\phi(s_t, a_t)^\top) \otimes I_d,$$
$$b_t = [\phi(s'_t, a'_t) \otimes I_d]C(x_t). \tag{68}$$

Recall the matrices $A$ and $C$ defined in Appendix D. For a constant $\xi > 0$, we define

$$H_t = \begin{bmatrix} A_t & B_t \\ \xi A_t & \xi C_t \end{bmatrix}, \qquad h_t = \begin{bmatrix} b_t \\ 0 \end{bmatrix}.$$

and

$$H = \begin{bmatrix} A & B \\ \xi A & \xi C \end{bmatrix}, \qquad h = \begin{bmatrix} b \\ 0 \end{bmatrix},$$

where $A = \mathbb{E}[A_t]$, $B = \mathbb{E}[B_t]$, $C = \mathbb{E}[C_t]$ and $b = \mathbb{E}[b_t]$.

For the matrix $H_t$, we have the following holds

$$\|H_t\|_F^2 = (1 + \xi^2) \|A_t\|_F^2 + \|B_t\|_F^2 + \xi^2 \|C_t\|_F^2$$
$$\leq (1 + \xi^2)[d^2 C_\phi^2 (C_m + 1)]^2 + d^2 C_\phi^2 C_m^2 + \xi^2 C_\phi^4 d^2. \tag{69}$$

which implies that $\|H_t\|_F \leq C_H$, where

$$C_H = \sqrt{(1 + \xi^2)[d^2 C_\phi^2 (C_m + 1)]^2 + d^2 C_\phi^2 C_m^2 + \xi^2 C_\phi^4 d^2}.$$

For the vector $h_t$, we can obtain $\|h_t\|_2 \leq C_h = dC_\phi R_C$ by following the steps similar to those for obtaining eq. (69).

The update in Algorithm 2 can be rewritten as

$$v_{t+1} = \Gamma_{R_v} (v_t + \alpha_t(H_t v_t + h_t)), \tag{70}$$

where $v_t = [\theta_t^\top, w_t^\top]^\top$, and $R_v = R_\theta \times \mathbb{R}^{Kd_g \times 1}$. Following the proof similar to those in Theorem 3 of Section 5.3.3 in [23], we can show that the matrix $H$ is Hurwitz under Assumption 4 and Assumption 5 with an appropriately chosen $\xi > \max\{0, -\mathrm{eig}_{\min}(C^{-1}[(A + A^\top)/2])\}$.

We define the following optimal point $v^* = [\bar{\theta}^{*\top}, w^{*\top}]^\top$ for the linear SA defined in eq. (70)

$$\langle \varphi(v^*), v - v^* \rangle \leq 0, \quad \forall v \in R_v,$$

where $\varphi(v) = Hv + b$. We also define $C_v = \|v^*\|_2$. It can be checked that there exist a positive constant $\lambda'_G$ such that

$$\langle \varphi(v^*) - \varphi(v), v^* - v \rangle \leq -\lambda'_G \|v - v^*\|_2^2. \tag{71}$$

We further define $\varphi(x_t, v) = H_t v + h_t$.

Following the steps similar to those for proving Lemma 7 and Lemma 8, we can obtain the following two lemmas.

**Lemma 11.** *Given a sample $(s_t, a_t, B_t, s'_t) \sim \mathcal{D}_d$ and $a'_t \sim \pi(\cdot|s'_t)$ and any $v \in R_v$, we have the following holds*

$$\|\varphi(x_t, v) - \varphi(v)\|_2^2 \leq 16C_H^2 \|\kappa - \kappa^*\|_2^2 + 16C^2 hPC_v^2 + 8C_h^2.$$

*Proof.* Based on the definition of $\varphi(x_t, v)$ and $\varphi(v)$, we can obtain the following

$$\|\varphi(x_t, v) - \varphi(v)\|_2^2 \leq 2 \|(H_t - H)v\|_2^2 + 2 \|h_t - h\|_2^2$$
$$\leq 4 \|(H_t - H)(v - v^*)\|_2^2 + 4 \|(H_t - H)v^*\|_2^2 + 2 \|h_t - h\|_2^2$$
$$\leq 16C_H^2 \|\kappa - \kappa^*\|_2^2 + 16C_h^2 C_v^2 + 8C_h^2.$$

$\square$

**Lemma 12.** *Consider the population GTD update $\varphi(v) = Hv + b$. We have*

$$\langle -\varphi(v), v - v^* \rangle \geq \lambda'_G \|v - v^*\|_2^2, \quad \forall v \in R_v.$$

We also have the following "three-point lemma" holds for the GTD update.

**Lemma 13.** *Consider the update of $w_t$ and $\theta_t$ in Algorithm 2. For all $v \in R_v$, we have the following holds*

$$-\alpha_t \langle \varphi(x_t, v_t), v_{t+1} - v \rangle + \frac{1}{2} \|v_{t+1} - v_t\|_2^2 \leq \frac{1}{2} \|v_t - v\|_2^2 - \frac{1}{2} \|v_{t+1} - v\|_2^2. \tag{72}$$

Using Lemma 13 and following the steps similar to those from eq. (45) to eq. (48), we can obtain

$$\frac{1}{2} \|v_t - v\|_2^2 - \frac{1}{2} \|v_{t+1} - v\|_2^2$$
$$\geq -\alpha_t \langle \varphi(v_{t+1}), v_{t+1} - v \rangle - \alpha_t \langle \varphi(x_t, v_t) - \varphi(v_t), v_t - v \rangle - \alpha_t^2 C_H^2 \|v_{t+1} - v\|_2^2$$
$$- \alpha_t^2 \|\varphi(x_t, \kappa_t) - \varphi(\kappa_t)\|_2^2. \tag{73}$$

Taking expectation on both sides of eq. (73), letting $v = v^*$, and using the fact that $-\langle \varphi(v_{t+1}), v_{t+1} - v^* \rangle \leq \lambda'_G \|v_{t+1} - v^*\|_2$ yield

$$(1 + 2\alpha_t \lambda'_G - 2\alpha_t^2 C_H^2) \mathbb{E}[\|v_{t+1} - v^*\|_2^2] \leq \mathbb{E}[\|v_t - v^*\|_2^2] + 2\alpha_t^2 \mathbb{E}[\|\varphi(x_t, \kappa_t) - \varphi(\kappa_t)\|_2^2]$$
$$\leq (1 + 32\alpha_t^2 C_H^2) \mathbb{E}[\|v_t - v^*\|_2^2] + +32C_h^2 C_v^2 + 16C_h^2, \tag{74}$$

where the second inequality follows from Lemma 12. Multiplying both sides of eq. (74) by $o_t$ and summing over iterations $t = 0, \cdots, T - 1$ yield

$$\sum_{t=0}^{T-1} a_t'' \mathbb{E}[\|v_{t+1} - v^*\|_2^2] \leq \sum_{t=0}^{T-1} b_t'' \mathbb{E}[\|v_t - v^*\|_2^2] + c'', \tag{75}$$

where

$$a_t'' = (1 + 2\alpha_t \lambda'_G - 2\alpha_t^2 C_H^2)o_t,$$
$$b_t'' = (1 + 32\alpha_t^2 C_H^2)o_t,$$
$$c'' = (32C^2 hPC_v^2 + 16C_h^2) \sum_{t=0}^{T-1} \alpha_t^2 o_t.$$

Now we let

$$o_t = (t + t_2)(t + t_2 + 1),$$
$$\alpha_t = \frac{4}{\lambda'_G(t + t_2 - 1)},$$
$$t_2 = \frac{34C_H^2}{\lambda_J^2} + 1.$$

Then, we can obtain the following

$$a_t'' - b_{t+1}'' = (1 + 2\alpha_t \lambda'_G - 2\alpha_t^2 C_H^2)s_t - (1 + 32\alpha_{t+1}^2 C_H^2)o_{t+1}$$
$$\geq (1 + 2\alpha_t \lambda'_G)o_t - (1 + 2\alpha_t^2 C_H^2 + 32\alpha_{t+1}^2 C_H^2)o_{t+1}$$
$$\geq (1 + 2\alpha_t \lambda'_G)o_t - (1 + 34\alpha_t^2 C_H^2)o_{t+1}$$
$$\geq (1 + 2\alpha_t \lambda'_G)o_t - (1 + \alpha_t \lambda'_G)o_{t+1}$$
$$\geq (t + t_2 + 1)\frac{(t + t_2 + 3)(t + t_2) - (t + t_2 + 2)^2}{t + t_2 - 1}$$
$$\geq 0,$$

where the second inequality follows from the fact that $\alpha_t \leq \frac{\lambda'_G}{34C_H^2}$.

Applying the above property to eq. (75) yields

$$a''_{T-1} t \mathbb{E}[\|v_T - v^*\|_2^2] \le b''_0 \|v_0 - v^*\|_2^2 + c'',$$

which implies

$$
\begin{aligned}
\mathbb{E}[\|v_T - v^*\|_2^2] &\le \frac{b''_0 \|v_0 - v^*\|_2^2}{a''_{T-1}} + \frac{c''}{a''_{T-1}} \\
&\le \frac{(1 + 16\alpha_0^2 C_H^2)(t_2 + 1)^2 \|v_0 - v^*\|_2^2}{(T + t_2 - 1)(T + t_2)} + \frac{128 C_h^2 C_v^2 + 64 C_h^2}{(T + t_2)\lambda_J^2}.
\end{aligned}
$$

Using the fact $\|\theta_T - \bar{\theta}^*\|_F^2 \le \|v_T - v^*\|_2^2$, we have

$$\mathbb{E}[\|\theta_T - \bar{\theta}^*\|_F^2] \le \frac{(1 + 16\alpha_0^2 C_H^2)(t_2 + 1)^2 \|v_0 - v^*\|_2^2}{(T + t_2 - 1)(T + t_2)} + \frac{128 C_h^2 C_v^2 + 64 C_h^2}{(T + t_2)\lambda_J^2},$$

which completes the proof.