# OpenReview forum: "A Unifying Framework of Off-Policy General Value Function Evaluation"
_NeurIPS.cc/2022/Conference — NeurIPS 2022 Accept_

### Official Review · Reviewer_5ZXZ · 2022-07-07

**Rating:** 7
**Confidence:** 2
**Soundness:** 3 good
**Presentation:** 2 fair
**Contribution:** 3 good

**Summary:**

The paper proposes a new off-policy algorithm to evaluate forward and backward general value functions that have the property of causal filtering. They show that the existing algorithm (GTD) fails in this (for example, failing to compute the ground truth GVF) and propose an algorithm to overcome its shortcomings.

Thanks to the authors for putting in the effort in doing this work!


**Questions:**

Questions:
- I’m a bit wary of the dependence that GenTD has on the function class being expressive enough given that performance degrades a lot if this assumption is not met (CFT vs. INCFT). Three subpoints: a) Even in the case presented in Section 5 (the CFT case), they don’t seem to achieve perfectly 0 error in the left graph of Figure 1. I’d imagine that in this relatively simplified setting, zero error could be achieved. b) The plots of INCFT suggest that performance drastically deteriorates when the assumption of expressiveness of the function class does not hold, which is a bit concerning since we wouldn't want an algorithm that has such a strong dependence on this function class expressiveness assumption. If performance degraded only slightly, then it would be another thing. c) It may also be the case that the CFT example used in the experiments is an extreme worst case. If that is so, then I recommend the authors vary the degree of failure of the assumption holding and see how sensitive GenTD is at different levels of the assumption holding true.

Suggestions:
- Title has a spelling mistake “evaluatio”
- For Figure 1, could the y-axis labels be made more clearer. The labeling of only 0 and 30 doesnt make it clear what exactly the errors are.
- The concept of GVF is relatively niche and so it must be made clear what GVFs actually are, for example in the intro. Its difficult to follow the intention of the work and what is exactly trying to be solved until Section 2 is reached.



**Limitations:**

No, they do not address it, but this seems to be more fundamental work with no direct societal impact.

**Strengths And Weaknesses:**

Strengths:
- I think this is an important area to research. GVFs are relatively under-researched and it's important to see this type of work.
- The 1) use of the concrete example on line 140 was helpful to communicate the idea of GVFs; 2) the comparisons between GenTD and GTD was also nice.

Weaknesses:
- Given that the concept of GVFs is unique and somewhat niche, I think the authors should be clearer on what GVFs exactly are (see suggestions below).
- The experimental section seems rather limited. The cited paper on GVFs General value function networks, 2021, seems to have a pretty detailed experimental section. It would be interesting to see how the proposed algorithm scales to harder domains.

---

> ### Author Response · Authors · 2022-08-02
> **Reply to Reviewer 5ZXZ**
>
> Many thanks for providing the helpful review.
>
> Q1: I’m a bit wary of the dependence that GenTD has on the function class being expressive enough given that performance degrades a lot if this assumption is not met (CFT vs. INCFT). Three subpoints: a) Even in the case presented in Section 5 (the CFT case), they don’t seem to achieve perfectly 0 error in the left graph of Figure 1. I’d imagine that in this relatively simplified setting, zero error could be achieved. b) The plots of INCFT suggest that performance drastically deteriorates when the assumption of expressiveness of the function class does not hold, which is a bit concerning since we wouldn't want an algorithm that has such a strong dependence on this function class expressiveness assumption. If performance degraded only slightly, then it would be another thing. c) It may also be the case that the CFT example used in the experiments is an extreme worst case. If that is so, then I recommend the authors vary the degree of failure of the assumption holding and see how sensitive GenTD is at different levels of the assumption holding true.
>
> A1: Regarding (a), in the left plot, our current experiment adopts a constant stepsize, which can converge only to a neighborhood of the optimal point due to non-vanishing variance of the update. It is expected that with diminishing stepsize, our algorithm should converge exactly to the optimal point. In Appendix A.1 in the revision, $\textbf{we have added an experiment}$, which demonstrates that under diminishing stepsize, our algorithm GenTD is able to achieve near zero bias error with CFT.
>
> Regarding (b), the INCFT we consider is rather a worse case, in which the one feature we remove from the CFT (to construct the INCFT) corresponds to a rather large entry of value function. Hence, it causes a large bias error. Alternatively, if we remove a feature corresponding to a smaller entry of value function from CFT to construct INCFT, then we expect the resulting bias error to be much smaller. In Appendix A.1 in the revision, $\textbf{we have added an experiment}$, in which a feature with small weight is removed to form an INCFT1. Figure 3 in Appendix A.1 demonstrates that the performance of GenTD decreases only slightly under such an INCFT compared with that with CFT.
>
> Regarding (c), Thanks for the suggestion. In Appendix A.1 in the revision, $\textbf{our experiment}$ compares the performance of GenTD under CFT, INCFT1 and INCFT2, where the expressive power of INCFT1 is higher than INCFT2. Our experiment (Figure 3) demonstrates that the bias error increases as the expressive power of the function approximation class decreases, which agrees with our theorem.
>
> Q2: Title has a spelling mistake “evaluation”
>
> A2: Thanks for pointing this out. We have fixed that in our revision.
>
> Q3: For Figure 1, could the y-axis labels be made more clearer. The labeling of only 0 and 30 doesn't make it clear what exactly the errors are.
>
> A3: Thanks for pointing it out. We have fixed that in our revision.
>
> Q4: The concept of GVF is relatively niche and so it must be made clear what GVFs actually are, for example in the intro. It's difficult to follow the intention of the work and what exactly is trying to be solved until Section 2 is reached.
>
> A4: Thanks for the suggestion! Since the mathematical definition of GVFs requires formal definition of MDP, it is a little difficult to define GVFs mathematically in the introduction. Instead, in the introduction, we describe the concept of GVFs at a high level, and refer the reader to later Sections 2.1 and 2.2 for formal mathematical definitions.

---

> > ### Comment · Reviewer_5ZXZ · 2022-08-05
> > **response to authors**
> >
> > Thank you for the response and addressing the concerns. I will raise the score. Its nice to see initial work in this very new area.

---

> > > ### Author Response · Authors · 2022-08-05
> > > **Many thanks**
> > >
> > > Many thanks for your further feedback and for raising the score! We really appreciate your efforts!

---

### Official Review · Reviewer_Ld7J · 2022-07-09

**Rating:** 5
**Confidence:** 2
**Soundness:** 4 excellent
**Presentation:** 3 good
**Contribution:** 3 good

**Summary:**

The paper considers evaluating multiple interrelated general value functions using offline data. A generalized TD learning algorithm is developed and its theoretical properties are analyzied in detail. The strengths and weaknesses of the paper are given in the next section. The contributions of the paper are mainly theoretical and include (1) establishment of the contraction property for both forward and backward value functions; (2) development of a generalized TD learning algorithm to overcome the limitations of previous baseline methods; (3) convergence analysis of the proposed algorithm; (4) identification of sufficient conditions under which the proposed algorithm would converge.

**Questions:**

Please refer to **Strengths and Weaknesses**

**Limitations:**

The main limitation includes the linearity assumption as well as a lack of extensive empirical studies. The former was partly discussed in the discussion where nonlinear function approximation was mentioned.

**Strengths And Weaknesses:**

Strengths:

1. The paper evaluates both forward and backward general value functions. Despite the richness of the literature on off-policy evaluation, these general value functions are less studied. In that sense, the paper targets an interesting research problem.
2. The proposed method considers multiple interrelated general value functions jointly rather than on a case-by-case basis.
3. A generalized TD learning algorithm is proposed to address the limitations of the gradient TD methods.
4. Convergence of the model parameter and the estimated value functions are investigated in detail.

Some comments:

1. There is a typo in the title of the pdf file. In addition, there is a question mark in the checklist.
2. The linearity assumption is quite strong. The algorithm might perform poorly in nonlinear systems with high-dimensional state information.
3. It remains unclear to me if the variance of "the reward to go" (mentioned on L23) can be represented in the form of the general value function. In particular, the variance of the cumulative reward would involve the interaction term that measures the covariance of the rewards at different time points.
4. Can you allow $B_j$ to be unobservable? The gradient of the value function (mentioned on L26) does not seem to be an observable quantity. It might be more useful to cover cases where $B_j$ needs to be estimated from the data as well.
5. The literature on off-policy evaluation with a general scalar value function is not thoroughly reviewed. In addition, it was mentioned on P2, L92 that these methods are not directly comparable. However, I do not agree with this argument. Suppose the initial state-action distribution concentrates on a particular state-action pair, then the value function is reduced to the state-action value. Combined with a kernel-type estimator, existing OPE methods such as (marginal) importance sampling, double robust estimation (double reinforcement learning) can be potentially applied to this setting. It remains unclear to me whether the proposed estimator is better.
6. The numerical study is oversimplified. It considers a simple toy example with 7 states and 2 actions. More extensive simulations based on e.g., OpenAI gym environments are needed to test the empirical performance of the proposed algorithm.

---

> ### Author Response · Authors · 2022-08-02
> **Reply to Reviewer Ld7J**
>
> Many thanks for providing the helpful review.
>
> Q1: There is a typo in the title of the pdf file. In addition, there is a question mark in the checklist.
>
> A1: Thanks! We have fixed those in our revision.
>
> Q2: The linearity assumption is quite strong. The algorithm might perform poorly in nonlinear systems with high-dimensional state information.
>
> A2: We adopt linear function approximation settings to simplify the technicality in theoretical development. Since there is no previous theoretical result on GVF evaluation, we take the first step to develop such a result in the linear function approximation setting. In fact, our GenTD can be generalized easily to nonlinear function approximation settings by incorporating some implementation techniques that were used in previous studies. Specifically, the density ratio $\rho$ in our GenTD can be estimated efficiently with neural network function approximation [1] and GVFs can be estimated accurately with neural networks by adopting the approximating scheme in [2]. Unlike the linear setting, in which density ratio and GVFs can be learned simultaneously, in the neural network approximation setting, we need to update two variables separately and design the learning rates carefully to make sure the algorithm still has convergence guarantee.
>
> [1] Zhang, R., Dai, B., Li, L., & Schuurmans, D. (2020). Gendice: Generalized offline estimation of stationary values. arXiv preprint arXiv:2002.09072
>
> [2] Comanici, G., Precup, D., Barreto, A., Toyama, D. K., Aygün, E., Hamel, P., ... & Mourad, S. (2018). Knowledge representation for reinforcement learning using general value functions.
>
> Q3: It remains unclear to me if the variance of "the reward to go" (mentioned on L23) can be represented in the form of the general value function. In particular, the variance of the cumulative reward would involve the interaction term that measures the covariance of the rewards at different time points.
>
> A3: The variance of “reward-to-go” can be captured by our forward GVFs framework. We provided a detailed discussion in Appendix B.1, in which we show that the variance of “reward-to-go” can be captured by the forward GVFs framework in Definiton 1.
>
> Q4: Can you allow $B_j$ to be unobservable? The gradient of the value function (mentioned on L26) does not seem to be an observable quantity. It might be more useful to cover cases where $B_j$ needs to be estimated from the data as well.
>
> A4: The general reward signal $B_j$ needs to be observable. For the evaluation of gradient of value function, $B$ takes the form of $B=[r,0]^\top$ (Please see the detailed discussion in Appendix B.1), which is clearly observable. More generally, for the GVFs satisfying the causal filtering property that we study in this paper, all the $B_j$ are observable, although some reward signals might be unobservable. When causal filtering does not hold, then $B_j$ could be unobservable and need to be estimated from data. In order to evaluate the GVF in such a more challenging setting, we possibly need to design multiple correlated GVFs evaluation processes, which is an interesting topic for future study.
>
> Q5: The literature on off-policy evaluation with a general scalar value function is not thoroughly reviewed.
>
> A5: Many thanks for the helpful comments! We agree that the distribution correction part of GenTD is related to OPE works. We have added the discussion of related work of OPE in Appendix B in our revision. We also agree that some OPE approaches may be applicable to the setting here. We also want to point out that GenDICE that we adopted in GenTD can have the following advantages compared with some of OPE methods such as importance sampling based approaches: (1) GenDICE is behavioral-policy agnostic, while importance sampling based approaches require the knowledge of behavior policy. (2) GenDICE suffers less from the variance compared with importance sampling approaches. Moreover, to our understanding of doubly robust estimation, it also needs to estimate the density ratio, and thus still needs to be combined with some density ratio estimation approaches in order to be useful.
>
> Q6: The numerical study is oversimplified. It considers a simple toy example with 7 states and 2 actions. More extensive simulations based on e.g OpenAI gym environments are needed to test the empirical performance of the proposed algorithm.
>
> A6: Thanks for the suggestion, we will add additional experiments in our revision.

---

### Official Review · Reviewer_MWog · 2022-07-11

**Rating:** 5
**Confidence:** 2
**Soundness:** 3 good
**Presentation:** 3 good
**Contribution:** 2 fair

**Summary:**

The problem formulation this paper is concerned with is the General Value Function (GVF) approximation task, where the algorithm's task is to estimate a vector which is defined as an expectation of some algebraic manipulation of the signals received. This vector is named a GVF.  On this task, the authors mainly compared their approach with another existing method called Gradient Temporal Difference (GTD). The main observation of this paper is that the goal of GTD is to minimize the empirical Mean Squared Projected Bellman Error (MSPBE), whereas a more legit goal would be the expected mean-squared projected Bellman error which they name as the Mean Squared Projected General Bellman Error (MSPGBE). To account for the shift from the empirical (data-dependent) mean to the expected mean, they applied an existing density ratio estimation method.

**Questions:**

I understand that I might be asking a question that might have been addressed by other papers, but again, why is density ratio the right thing to estimate? There are other types of divergences too, is there something special about RL that makes density ratio easy to estimate?

Could you explain why for the counter-example you proposed for GTD, why would GenTD work? Or is the counter-example merely evidence that GTD is not perfect?

**Limitations:**

I guess assumptions count for limitations. A better limitation from my point of view would be to specify for which interesting classes of RL tasks, the algorithm GenTD wouldn't work.

**Strengths And Weaknesses:**

Strengths:\
    Originality: The motivation is legit.\
    Quality: The logic chain works through, it is clear what they are doing. \
    Clarity: The story is easy to read. (I didn't check any proof.)\


Weakness:\
    Significance: \
        1. Seems A+B, where A = GTD, B = density ratio estimation.\
        2. More precisely, I put this as a weakness because personally, I don't quite get the logic why learning density ratio would be more accurate. The authors put as a counter-example where GTD fails for Complete F_\Phi. However, if for the same example, the density ratio is not learned very well, why would GenTD work?

---

> ### Author Response · Authors · 2022-08-02
> **Reply to Reviewer MWog**
>
> Many thanks for providing the helpful review.
>
> Q1: Seems A+B, where A = GTD, B = density ratio estimation.
>
> A1: We respectfully disagree with the reviewer. We clarify that our GenTD can be viewed as **TD + distribution correction**, not **GTD + distribution correction**. Hence, the nature of GenTD is very different from GTD. To elaborate further, **GTD** adopts an objective function that takes the average over the behavior sampling distribution. As a result, GTD could suffer from large bias error even when the expressive power of the approximation function class is very large or complete (as we demonstrate in our counter-example). In contrast, our **GenTD** corrects the behavior sampling distribution to the target stationary distribution via learning the density ratio, and is thus guaranteed to converge to the ground truth value functions as long as the expressive power of the approximation function class is sufficiently large. Note that such an advantage of GenTD over GTD is demonstrated by our counterexample.
>
> In fact, the main contribution of the paper is to study the general value function (GVF) evaluation problem, i.e., evaluating $\textbf{multi-dimensional value functions with correlations}$, not the $\textbf{scalar}$ value function evaluation problem for which TD/GTD was typically studied in the literature. It is a not trivial fact whether applying GenTD to such a much more challenging GVF problem can have good performance or guaranteed convergence. In fact, it can diverge for arbitrary GVF problems. The contribution of this paper is to show that many GVF problems encountered in RL practice do satisfy a causal filtering structure, which guarantees that GenTD can be applied with provable convergence.
>
> Q2: More precisely, I put this as a weakness because personally, I don't quite get the logic why learning density ratio would be more accurate. The authors put as a counter-example where GTD fails for Complete $F_\Phi$. However, if for the same example, the density ratio is not learned very well, why would GenTD work?
>
> A2: See our answer to Q4.
>
> Q3: I understand that I might be asking a question that might have been addressed by other papers, but again, why is density ratio the right thing to estimate? There are other types of divergences too, is there something special about RL that makes density ratio easy to estimate?
>
> A3: In the off-policy setting, correcting the distribution mismatch in the objective function naturally yields the quantity of the density ratio to be learned, e.g., $E_\nu[F(x)] = E_d[\frac{\nu(x)}{d(x)} F(x)] = E_d[\rho(x) F(x)]$, where $\nu(x)$ is visitation distribution (target) and $d(x)$ is sampling distribution (behavior). Other types of divergences don’t serve such an exact mathematical purpose here.
>
> Q4: Could you explain why for the counterexample you proposed for GTD, why would GenTD work? Or is the counter-example merely evidence that GTD is not perfect?
>
> A4: In our A1 to Q1, we explain the central difference between the design of our GenTD and GTD, which justifies the advantage of GenTD over GTD. The counter-example further illustrates such an advantage. In the counterexample, GenTD is guaranteed to converge to the ground truth value function with 0 error, while GTD suffers from very large bias error. The reason is that, when the function approximation class is complete (which is the setting of our counterexample), learning density ratio using GenDICE (as in GenTD) does not have bias error, and the optimization error can be guaranteed to be arbitrarily small due to the convergence guarantee in [1]. With this accurately learned density ratio, GenTD can correct the distribution mismatch completely and converge to the ground true value function (as we establish in Theorem 1 in this paper). In contrast, GTD does not correct the distribution mismatch even with complete function approximation class (due to the nature of its design), thus still suffering from large bias error even when the function approximation class is complete.
>
> Q5: Specify for which interesting classes of RL tasks, the algorithm GenTD wouldn't work.
>
> A5: When the problem doesn’t satisfy the causal filtering structure, our GenTD does not have a convergence guarantee. A more complicated algorithm design is required in order to handle this more challenging setting.
>
> Finally, we thank the reviewer again for the helpful comments for our work. If our response resolves your concerns to a satisfactory level, we kindly ask the reviewer to consider raising the rating of our work. Certainly, we are more than happy to address any further questions you may have during the discussion period.

---

> ### Author Response · Authors · 2022-08-05
> **Could you please check our response**
>
> Dear Reviewer MWog,
>
> Since the author-reviewer discussion period has started for a few days, we will appreciate if you could check our response to your review comments soon. This way, if you have further questions and comments, we can still reply before the author-reviewer discussion period ends. If our response resolves your concerns, we kindly ask you to consider raising the rating of our work. Thank you very much for your time and efforts!

---

> ### Author Response · Authors · 2022-08-08
> **Your prompt response is highly appreciated**
>
> Dear Reviewer MWog:
>
> As the author-reviewer discussion period ends soon, we will appreciate very much if your could check our response promptly. In particular, our response has explained in detail about your questions on the algorithm (Q1), the density ratio (Q3), and counter-example (Q2/Q4). We also clarified our main contribution in the answer to Q1. If our response resolves your concerns, we kindly ask you to consider raising the rating of our work. We are also more than happy to answer your further questions. Thank you very much for your time and efforts!

---

> ### Comment · Reviewer_MWog · 2022-08-09
> **Reply to the rebuttal**
>
> Thanks for the rebuttal and sorry for the late reply. You have addressed many of my concerns. I can see that you have put thoughts into the approach so I will increase my score.

---

> > ### Author Response · Authors · 2022-08-09
> > **Many thanks**
> >
> > We thank the reviewer very much for the further feedback and for raising the score!

---

### Official Review · Reviewer_bpnm · 2022-07-12

**Rating:** 6
**Confidence:** 3
**Soundness:** 4 excellent
**Presentation:** 4 excellent
**Contribution:** 3 good

**Summary:**

This paper proposes a new method called GenTD for off-policy evaluation of GVFs. The algorithm can further estimate multiple intercorrelated GVFs at once under some reasonable (causal) assumptions. The contributions are mainly theoretical (convergence proofs). Some empirical evaluation on a small MDP is provided.

**Questions:**

- L259 could you comment on the choice and importance of psi?
- L260, you assume the matrix A to be non-singular, however if psi' is intended to be P_pi \Psi, then shouldn't this matrix be singular because it contains I - P_\pi which is singular?
- I am puzzled by the strength of the convergence results you have using only A1 and A2. For instance, why doesn't it suffer from deadly-triad-like issues? I had expected to see some assumptions on the spectrum of some matrices related to phi, P and D.

Minor:
- What is R_\theta L241? I may have missed the definition
- I may also have missed the definition/intuition behind m(x) L251
- I may also have missed the definition of the F-norm L296

**Limitations:**

Limitations are mentioned above. Beyond the questions on the theory, the limitations appear most on the practical side. In the paper the empirical evaluation may be lacking and thus making it harder to convince the community that this algorithm would be a safe and efficient one. More broadly, it is not clear to me how this algorithm would be extended to the non-linear setting.

**Strengths And Weaknesses:**

# Strengths
Overall, the paper tackles an interesting problem, identifies a shortcoming of previous algorithms and proposes a solution with theoretical guarantees. I found it overall pretty clear.

# Weaknesses

## Scalability
I have some concerns about the scalability of the method and whether it could extent beyond the linear regime. In particular, Alg 1 requires some projections step that may not make sense in the non-linear regime. Could authors comment on whether they think it is an issue?

## Experiments
The experimental validation is weak in my opinion. I understand this is a theory paper and the main contribution is the algorithm and its guarantees, however I believe it is also in the authors interest to showcase that the algorithm performs well on environments more complex. In particular, GenDICE which this work builds upon had some empirical evaluation on non-trivial gridworlds as well as a simple control (Half-Cheetah) environment.
As far as I know, methods based on min-max problems (DualDICE etc) are known to be quite unstable, so I think it is important that you showcase the robustness of your algorithm.

---

> ### Author Response · Authors · 2022-08-02
> **Reply to Reviewer bpnm**
>
> Many thanks for providing the helpful review.
>
> Q1: I have some concerns about the scalability of the method and whether it could extend beyond the linear regime. In particular, Alg 1 requires some projection steps that may not make sense in the non-linear regime. Could authors comment on whether they think it is an issue?
>
> A1: Great question! In the nonlinear function approximation setting, projection can be difficult as the reviewer mentioned. Instead, a practical way to ensure the boundedness of the parameters is to use an $l_2$ regularizer so that the parameter does not blow up during the training.
>
> Q2: L259 could you comment on the choice and importance of $\psi$?
>
> A2: $\psi$ should be chosen carefully so that the matrix $A$ is non-singular to guarantee the convergence of GenTD. Moreover, we also want to make sure that each entry of the feature vector $\psi$ is not too large so that the linear function approximation based on $\psi$ can well approximate the density ratio in a reasonable region.
>
> Q3: L260, you assume the matrix $A$ to be non-singular, however if $\psi$' is intended to be $P_pi \Psi$, then shouldn't this matrix be singular because it contains $I - P_\pi$ which is singular?
>
> A3: We assume that the reviewer asks about the tabular setting (which is not considered in our paper). In such a tabular setting, indeed, matrix $A$ is not guaranteed to be non-singular. As a typical solution, we can add a regularization to ensure the non-singularity of the problem and hence the convergence of the algorithm. However, in the linear function approximation setting (that we consider in our paper), it is reasonable to expect that the careful design of the feature $\psi$ can ensure $A$ is non-singular.
>
> Q4: I am puzzled by the strength of the convergence results you have using only A1 and A2. For instance, why doesn't it suffer from deadly-triad-like issues? I had expected to see some assumptions on the spectrum of some matrices related to $\Phi, P$ and $D$.
>
> A4: We need to assume the feature matrix $\Phi$ has linear independent columns (see line 203), which we will make as an explicit assumption in our revision. For the transition kernel $P$, we require it to have stationary distribution (see line 117). For the density ratio matrix $D$, we require each entry to be lower-bounded (see Assumption 1).
>
> Q5: What is $R_\theta$ L241? I may have missed the definition
>
> A5: $R_\theta$ is the projection radius of $\theta$ defined in Alg 1.
>
> Q6: I may also have missed the definition/intuition behind m(x) L251
>
> A6: $m$ and $\hat{m}$ are coefficient matrices such that the TD error expressions hold. In the simple scalar setting (with a single value function), clearly $m=\hat{m}=\gamma$. Intuitively, they capture the correlations between difference estimations in forward and backward GVFs evaluation settings, respectively.
>
> Q7: I may also have missed the definition of the F-norm L296
>
> A7: F-norm is the Frobenius-norm.
>
> Q8: It is not clear to me how this algorithm would be extended to the non-linear setting.
>
> A8: In fact, our GenTD can be generalized easily to nonlinear function approximation settings by incorporating some implementation techniques that were used in previous studies. Specifically, the density ratio $\rho$ in our GenTD can be estimated efficiently with neural network function approximation [1] and GVFs can be estimated accurately with neural networks by adopting the approximating scheme in [2]. Unlike the linear setting, in which density ratio and GVFs can be learned simultaneously, in the neural network approximation setting, we need to update two variables separately and design the learning rates carefully to make sure the algorithm still has convergence guarantee.
>
>
> [1] Zhang, R., Dai, B., Li, L., & Schuurmans, D. (2020). Gendice: Generalized offline estimation of stationary values. arXiv preprint arXiv:2002.09072
>
> [2] Comanici, G., Precup, D., Barreto, A., Toyama, D. K., Aygün, E., Hamel, P., ... & Mourad, S. (2018). Knowledge representation for reinforcement learning using general value functions.

---

> > ### Comment · Reviewer_bpnm · 2022-08-05
> > **reply**
> >
> > Thank you for your answer and precisions.
> >
> > Q1: Yes so of course that's a possibility but it doesn't work well usually with neural networks as you can have expressive functions with relatively low weights. Of course there are other (and better!) regularization methods, but I'm a bit wary of the stability these xDICE based methods from experience. The fact that you only perform evaluation in a simple MDP with a few states does not really convince me this will be a practical method that will be used.
> >
> > Q3: Maybe I misundertood something. Is $\psi' = P^\pi \psi$? In that case, as $P^\pi$ has an eigenvalue of 1 associated with the left and right vectors $d^\pi$ and $\mathbb{1}$ (all ones), $I - P^\pi$ is singular, therefore, it doesn't seem so unlikely for $A$ to be singular no?
> >
> > Q4: Ok so reading more closely it seems your method is more a generalization of gradient TD than TD no? You still multiply on the left by $P^\pi \Phi$.
> > In that case, if I understood correctly, could you
> > 1) comment on why you don't need double sampling, ie have independent samples of $s', a'$ for $\phi$ and $\delta$ in  $g$?
> > 2) Methods based on the Bellman residual are known to be usually much slower than TD. What happens if you try to estimate the value function compared to TD?
> >
> > Q5: Ok! I think it should be defined in the main text as it is not obvious to look for a definition in the algorithms.
> > Q7: probably should be mentioned in the text if I missed it
> >
> > I will wait until the reviewer/AC discussion to do any update to my score, thank you again for the precisions!

---

> > > ### Author Response · Authors · 2022-08-07
> > > **Thanks for your reply**
> > >
> > > We thank the reviewer for carefully reviewing our revision and response. Below is our response.
> > >
> > > Q1: I'm a bit wary of the stability of these xDICE based methods from experience. The fact that you only perform evaluation in a simple MDP with a few states does not really convince me this will be a practical method that will be used.
> > >
> > > A1: Many thanks for the comment. We generally agree with the reviewer that making these approaches work in practice will still take significant efforts, possibly adding various additional tricks, although GenDICE has achieved initial empirical success in OPE by adjusting the distribution mismatch with a batch of samples.
> > >
> > > We also want to call the reviewer’s attention that our main focus here is on the general value function (GVF) evaluation problem, i.e., evaluating $\textbf{multi-dimensional value functions with correlations}$, which has not been formally studied in an off-policy setting. It is a non-trivial fact whether applying any off-policy TD method to such a challenging GVF problem can have good performance or guaranteed convergence. In fact, it can diverge for arbitrary GVF problems. The contribution of this paper is to show that many GVF problems encountered in RL practice do satisfy a causal filtering structure, which guarantees that a well designed off-policy TD method (such as GenTD) can be applied with provable convergence. With such an initial progress, we hope to motivate more and better designs for such a type of GVF problems.
> > >
> > > Q3: $I-P^{\pi\top}$ is singular. Should it be likely for A to be singular?
> > >
> > > A3: Thanks for the question. The definition of A is $A = \Psi^\top(I-P_\pi^{\top})D\Psi$. As reviewer mentioned, the middle matrix $I-P_\pi^{\top}$ in $A$ has zero eigenvalue, and is singular. However, whether $A$ is singular depends on whether the columns of $\Psi$ include any  eigenvector corresponding to zero eigenvalue, namely, whether $\Psi$ has nonzero projection in the null space of the middle matrix $I-P_\pi^{\top}$. If so, $A$ can be singular. On the other hand, if $\Psi$ does not intersect with the null space of $I-P_\pi^{\top}$ (which can be up to our design), then $A$ is nonsingular. In our analysis, we simplify the problem by assuming $A$ is non-singular. But for a singular matrix $A$, we can add an $l_2$ regularization to ensure the nonsingularity and then all our analysis will still hold.
> > >
> > > Q4: (1) Your method is more a generalization of gradient TD than TD?
> > >
> > > A4: (1) Many thanks for the question! In fact, our approach GenTD can be viewed as “TD learning + distribution correction”, with adaptation to handle updates of multiple correlated GVFs. The nature of GenTD is very different from Gradient TD (GTD).
> > >
> > > To elaborate further, GTD adopts an objective function that takes the average over the behavior sampling distribution $P_D(s,a) = D(s)\pi_b(a|s)$, where $\pi_b(a|s)$ is the behavior policy. In its design, GTD corrects only the mismatch between behavior and target policy, via the policy ratio $\pi(a|s)/\pi_b(a|s)$ (where $\pi(a|s)$ is target policy), but doesn’t correct the distribution mismatch of $D(s)$. This is not sufficient, because we not only need to correct the policy but also need to correct the behavior sampling distribution $D(s)$.
> > > In contrast, our GenTD correct the entire $P_D(s,a)=D(s)\pi_b(a|s)$ to $P_\pi(s,a) = \mu_\pi(s)\pi(a|s)$ (where $\mu_\pi(s)$ is the stationary distribution under target policy $\pi$) via learning the density ratio $\rho(s,a) = P_\pi(a|s)/P_D(s,a)$. In this way, GenTD corrects both $D(s)$ and $\pi_b(a|s)$, and thus has better performance guarantee than GTD.
> > >
> > > Q4: (2) Why don't you need double sampling for $(s^\prime, a^\prime)$ in $\phi$ and $\delta$ in $g$?
> > >
> > > A4: (2) First recall that the updates of our GenTD and GTD are given as follows:
> > >
> > > GenTD: update= $E_\pi[\phi\delta]$
> > >
> > > GTD: update = $E_D[(\phi - \gamma\phi^\prime)\phi^\top] E_D[\phi\phi^\top]^{-1} E_D[\phi\delta]$
> > >
> > > From the above equations, it can be seen that GenTD does not have a double sampling issue, because its update expression does not have a product of two expectations, whereas the update of GTD does have the form of two expectations multiplied together and hence requires double sampling.
> > >
> > > Q4: (3) Methods based on the Bellman residual are known to be slower than TD.
> > >
> > > A4: (3) We agree that GTD does suffer from slow convergence issues caused by Bellman residuals. However, our GenTD is essentially “TD learning + distribution correction”, where the design is not based on the Bellman residual. Thus, GenTD does not suffer from the slow convergence issues that are encountered by GTD-type algorithms.
> > >
> > > We thank the reviewer again for your efforts. We understand that the reviewer will wait until the reviewer/AC discussion to do the score update. We would like to kindly ask the reviewer to support our paper as an expert on this topic if our response fully addresses your questions. Certainly, we will be more than happy to answer your further questions.

---

> > > > ### Comment · Reviewer_bpnm · 2022-08-08
> > > > **Reply**
> > > >
> > > > Thank you for your technical replies, especially concerning the distinction GTD / GenTD, this clarifies things for me!
> > > > I will raise the points you made in the discussion.

---

> > > > > ### Author Response · Authors · 2022-08-08
> > > > > **Thanks again**
> > > > >
> > > > > Many thanks for the further feedback. We also appreciate the efforts and the time that the reviewer will spend during the future discussion period.

---

### Author Response · Authors · 2022-08-09
**Many thanks to All Reviewers and ACs**

We truly thank all reviewers’ insightful and constructive comments, which helped significantly to improve our paper. We also thank the ACs for their time and efforts during the review process.

---

### Meta-Review · Area_Chair_dbHg · 2022-08-27

**Recommendation:** Accept
**Confidence:** Less certain

**Metareview:**

The paper proposes a new algorithm called GenTD for the estimation of multiple general value functions (predictive and retrospective) from off-policy data. The paper shows convergence guarantees for this algorithm to the ground truth for a certain class of general value functions with causal filtering.

The initial reviews were mixed. On the positive side, the reviewers found the writing to be clear overall, found the studied problem important and appreciated the theoretical results. On the negative side, several reviewers voiced concerns regarding the experimental evaluation. Other concerns are the limitation of the linear setting and possible extensions to the non-linear setting as well as the significance, specifically, whether this work is merely a combination GTD and density ratio estimation. The authors' response could alleviate these concerns, further clarifying the contributions of the paper as well as adding additional experimental results. After the discussion with the authors, all reviewers view the paper positively and the AC agrees. All in all, this paper is recommended to be accepted.

**Award:**

No

---

### Decision · Program_Chairs · 2022-09-14

Accept